

**Greenhouse gas emissions and reactive nitrogen releases from rice production**
**with simultaneous incorporation of wheat straw and nitrogen fertilizer**
Longlong Xia[a,b], Yongqiu Xia[a], Shutan Ma[a,b], Jinyang Wang[a], Shuwei Wang[a,b], Wei
Zhou[a,b], Xiaoyuan Yan [a*]
[a.] State Key Laboratory of Soil and Sustainable Agriculture, Institute of Soil Science,
Chinese Academy of Sciences, Nanjing 210008, China.
[b.] University of Chinese Academy of Sciences, Beijing 100049, China.
[*]**Corresponding author**: Xiaoyuan Yan
State Key Laboratory of Soil and Sustainable Agriculture, Institute of Soil Science,
Chinese Academy of Sciences, Nanjing 210008, P. R. China
Phone number: +86 025 86881530, Fax: +86 025 86881000
Email address: yanxy@issas.ac.cn



## Abstract

The impacts of simultaneous inputs of crop straw and nitrogen (N) fertilizer on greenhouse gas (GHG) emissions and reactive nitrogen (Nr) releases from rice production in intensive agricultural regions are not well understood. A field experiment was established in a rice–wheat cropping system in the Taihu Lake region (TLR) of China since 2013 to evaluate the GHG intensity (GHGI), Nr intensity (NrI) and environmental costs of concurrent inputs of wheat straw and N fertilizer to rice paddies. The field experiment included five treatments of different N fertilization rates for rice production: 0 (RN0), 120 (RN120), 180 (RN180), 240 (RN240) and 300 kg N ha$^{-1}$ (RN300, traditional N applied rate in the TLR). Wheat straws were fully incorporated into soil before rice transplantation in all treatments. The results showed that the response of rice yield to N application rate successfully fitted a quadratic model. Nitrous oxide (N$_2$O) emissions were increased exponentially as N fertilization rates increased, while methane (CH$_4$) emissions increased slightly with wheat straw rates increased. The estimated soil organic carbon sequestration rate varied from 129.58 (RN0) to 196.87 kg C ha$^{-1}$ yr$^{-1}$ (RN300). Seasonal average GHGI of rice production ranged from 1.20 (RN240) to 1.61 kg CO$_2$-equivalent (CO$_2$-eq) kg$^{-1}$ (RN0), while NrI varied from 2.14 (RN0) to 10.92 g N kg$^{-1}$ (RN300). CH$_4$ emissions dominated GHGI with proportion of 70.2-88.6%, while ammonia (NH$_3$) volatilization dominated NrI with proportion of 53.5-57.4% in all fertilization treatments. The damage costs to environment incurred by GHG and Nr releases from current rice production (RN300) accounted for 8.8% and 4.9% of farmer's incomes, respectively. Cutting the traditional application rate of N fertilizer from 300 to 240 kg N ha$^{-1}$ improved rice yield and nitrogen use efficiency by 2.14% and 10.30%, respectively, whilst simultaneously reduced GHGI by 13%, NrI by 23% and total environmental costs by 16%.



Moreover, the reduction of 60 kg N ha$^{-1}$ improved farmer's income by 639 ¥ ha$^{-1}$, which would
provide them with an incentive to change their traditional N application rate. Our study suggests
that GHG and Nr releases, especially the CH$_4$ emission and NH$_3$ volatilization, from rice
production in the TLR could be further curbed, considering the current incorporation pattern of
straw and N fertilizer.
Key words: Taihu Lake region, greenhouse gas intensity, Nr intensity, rice production, straw
incorporation



















## 1 Introduction


Rice is the staple food for the majority of the world's population. However, while
industriously feeding the world's population, rice production is an important source of greenhouse
gas (GHG) emissions and reactive nitrogen (Nr) releases (Yan et al., 2009; Chen et al., 2014).
Rice production in China involves heavy methane ($CH_4$) emissions due to current water regime
management and straw incorporation practices (Yan et al., 2009). Besides, the lower nitrogen use
efficiency for rice cultivation in China (approximately 31%) aggravates the release of various Nr
species, thus threatening ecosystem functions (Galloway et al., 2008; Zhang et al., 2012). Such a
dilemma highlights the need for the simultaneous evaluation of GHG emissions and Nr losses for
rice production in China. And rice cultivation in intensive agricultural regions, characterized by
high inputs of N fertilizer and crop residues, should be prioritized for the implementation of such
evaluation (Ju et al., 2009; Chen et al., 2014).
Taihu Lake region (TLR) is one of the most productive areas for rice production in China,
largely owing to the popularity of intensive cultivation (Zhao et al., 2012a; Zhao et al., 2012b).
Currently, rice yield of this region in some fields can reach up to 8000 kg $ha^{-1}$ or even higher (Ma
et al., 2013; Zhao et al., 2015). However, these grain yields are achieved with a cost to
environment (Ju et al., 2009). TLR generally receives 550-600 kg N $ha^{-1}$ $yr^{-1}$, with the
rice-growing season accounting for nearly 300 kg N $ha^{-1}$ (Zhao et al., 2012b). Asides from these
excessive N inputs, TLR also experiences high amounts of crop residue incorporation, which is
highly encouraged by local governments (Xia et al., 2014). However, direct straw incorporation
before rice transplantation triggers substantial $CH_4$ emissions (Ma et al., 2009; Ma et al., 2013).
Besides such substantial releases of Nr and GHGs in a direct way, indirect releases during the



production of various agricultural materials used for farming operations in the TLR, are also not
ignorable, due to higher input rates of these materials caused by intensive cultivation (Zhang et al.,
2013; Cheng et al., 2014). This warrants the need for life-cycle assessment of GHG emissions and
Nr releases with respect to rice production in this region.

Considerable environmental costs can be caused by the direct and indirect releases of GHGs

and Nr from rice production in the TLR, for instance, in the form of global warming, water
eutrophication, or soil acidification (Ju et al., 2009; Xia and Yan, 2011; Xia and Yan, 2012).
Previous studies have proven that environmental costs assessment could provide guidance for
emerging policy priorities in mitigating certain GHG or Nr species, after quantifying both their
release amounts and damage costs to ecosystems (Gu et al., 2012). However, the life-cycle
assessment of total GHG and Nr releases, and the environmental costs they incur from rice
production in the TLR under the current conditions of high inputs of N fertilizer and crop straw,
are scarce.

In the present study, we conducted two years of simultaneous measurements of $CH_4$ and $N_2O$

emissions from a rice-wheat cropping system in the TLR to evaluate the impacts of simultaneous
inputs of crop straw and N fertilizer on (1) net global warming potential (NGWP) and GHG
intensity (GHGI), (2) total Nr losses and Nr intensity (NrI), (3) environmental costs incurred by
GHG and Nr releases of rice production, from perspective of life-cycle assessment.
**2 Materials and methods**
**2.1 Experimental site**

The field experiment was conducted in a paddy rice field at Changshu Agroecological

Experimental Station (31°32′93″N, 120°41′88″E) in Jiangsu province, which is located in the TLR





of China where the cropping system is primarily dominated by summer rice (*Oryza sativa* L.,) and
winter wheat rotation. The climate of the study area is subtropical monsoon, with a mean air
temperature of 16.1°C and mean annual precipitation of 990 mm, of which 60-70% occurs during
the rice-growing season. The daily mean temperature and precipitation during two rice-growing
seasons from 2013 to 2014 are shown in Fig.1. The paddy soil is classified as an Anthrosol, which
develops from lacustrine sediments. The topsoil (0-20cm) has a pH of 7.68 ($H_2O$). The bulk
density is 1.16 g $cm^{-3}$, the organic C content is 20.1 g C $kg^{-1}$, the total N is 1.98 g $kg^{-1}$, the
available P is 11.83 mg $kg^{-1}$ and the available K is 126 mg $kg^{-1}$.
**2.2 Experimental design and field management**

The field experiment included five treatments of different N fertilization rates for rice

production: 0 (RN0), 120 (RN120), 180 (RN180), 240 (RN240) and 300 kg N $ha^{-1}$ (RN300,
traditional N applied rate in the TLR). Consistent with local practices, wheat straws were
harvested, chopped and fully incorporated into soil before rice transplantation in all treatments
(Table 1). All of the treatments are laid out in a randomized block design with three replicates, and
each plot covered an area of 3 m × 11 m (33 $m^2$).

Rice is transplanted in the middle of June and harvested at the beginning of November. N

fertilizer (in the form of urea) was split into three parts during the rice-growing season: 40% as
basal fertilizer; 30% as tillering fertilizer; and 30% as panicle fertilizer. Phosphorus (in the form of
calcium superphosphate) and potassium (in the form of potassium chloride) were applied as basal
fertilizer at rates of 30 kg $P_2O_5$ $ha^{-1}$ and 60 kg $K_2O$ $ha^{-1}$, respectively. All basal fertilizers were
thoroughly incorporated into the soil through plowing, while topdressing fertilizers were applied
evenly to the soil surface. According to local practices, the water regime of 'flooding-midseason



drainage-flooding-moist but non-waterlogged by intermittent irrigation' was adopted. Details of
the specific agricultural management practices for rice production are provided in Table 1.
**2.3 Gas fluxes and topsoil organic carbon sequestration rate**
The $CH_4$ and nitrous oxide ($N_2O$) fluxes during the rice-growing seasons of 2013 and 2014
were measured using a static chamber and gas chromatography technique. Details of the
procedures used for sampling and analysis the gases were described in Xia et al. (2014).
Considering the fact that the soil organic carbon sequestration rate (SOCSR) of this
short-term field experiment could not be measured directly, we used the following relationship
between the straw input rate (kg C $ha^{-1}$ $yr^{-1}$) and SOCSR (kg C $ha^{-1}$ $yr^{-1}$), obtained via an
on-going long-term straw application experiment in the same region, to calculate the SOCSR in
this study:
SOCSR = Straw input rate × 0.0603 + 31.39 ($R^2$ = 0.92);          (1)
This long-term field experiment is also taking place at the Changshu Agroecological
Experimental Station (since 1990), which includes three straw application levels: 0, 4.5 t, and 9.0 t
dry-weight $ha^{-1}$ $yr^{-1}$ and the N application rate for rice cultivation in these treatments is 180 kg N
$ha^{-1}$. The estimated SOCSR (from 1990 to 2012) for these three treatments was 10.65, 194.96 and
254.83 kg C $ha^{-1}$ $yr^{-1}$ (Xia et al., 2014). The equation (1) was established based on above straw
input rates and the estimated SOCSR. We used the average straw input rates of the two
rice-growing seasons to estimate the SOCSR. The on-going long-term experiment and the
experiment in this study received similar agricultural managements. Details of the on-going
long-term experiment are described in Xia et al. (2014).
**2.4 Net global warming potential and greenhouse gas intensity**



The net global warming potential (NGWP, kg $CO_2$ eq $ha^{-1}$) and greenhouse gas intensity
(GHGI, kg $CO_2$ eq $kg^{-1}$) of rice production in the TLR was calculated using the following
equations:

$$NGWP = \sum_{i=1}^{m} AI_{ico_2} + CH_4 \times 25 + N_2O \times 44/28 \times 298 - SOCSR \times 44/12; \qquad (2)$$

$$GHGI = NGWP/rice\ yield; \qquad (3)$$

Here, $AI_{ico_2}$ denotes the GHG emissions from the production and transportation of agricultural
inputs, which are calculated by multiplying their application rates by their individual GHG
emission factors, such as synthetic fertilizers, diesel oil, electricity and pesticides (Liang, 2009;
Zhang et al., 2013). $CH_4$ (kg $CH_4$ $ha^{-1}$), $N_2O$ (kg N $ha^{-1}$) and SOCSR (kg C $ha^{-1}$ $yr^{-1}$) represent
the $CH_4$ emissions and $N_2O$ emissions from rice production, and the SOC sequestration rate,
respectively.
**2.5 Total Nr losses and Nr intensity**

The total Nr losses (kg N $ha^{-1}$) and Nr intensity (NrI, g N $kg^{-1}$) were calculated using the

following equations:

$$Total\ Nr\ losses = \sum_{i=1}^{m} AI_{iN_r} + (NH_3 + N_2O + N_{Leaching} + N_{Runoff})_{rice}; \qquad (4)$$

$$NH_3\ volatilization = 0.17 \times N\ fertilizer\ rate + 0.64; \qquad (5)$$

$$N\ runoff = 5.39 \times exp\ (0.0054 \times N\ fertilizer\ rate); \qquad (6)$$

$$N\ leaching = 1.44 \times exp\ (0.0037 \times N\ fertilizer\ rate); \qquad (7)$$

$$NrI = (1000 \times Total\ Nr\ losses)\ /rice\ yield\ ; \qquad (8)$$

Here, $AI_{iNr}$ denotes the Nr lost (mainly through $N_2O$ and $NO_X$ emissions) from the production
and transportation of agricultural inputs (Liang, 2009; Zhang et al., 2013), while
'$(NH_3+N_2O+N_{Leaching}+N_{Runoff})_{rice}$' represents the $NH_3$ volatilization, $N_2O$ emissions, N leaching



and runoff during the rice-growing season. We conducted a meta-analysis of published literature to
establish Nr empirical models to stimulate the Nr losses, such as $NH_3$ volatilization (Equation 5),
N leaching and runoff (Equation 6 and 7), from different treatments. Specific details regarding this
literature survey are provided in Appendix A.
**2.6 Total environmental costs incurred by GHG and Nr releases and farmer's**
**income**

The total environmental costs (¥ ha$^{-1}$) incurred by GHG and Nr releases and farmer's income

from rice production in the TLR was calculated based on the following equations:

$$Environmental\ costs = \sum_{i=1}^{n}(Nr_iA \times DC_i) + CO_2A \times DC_{CO2};    (9)$$

Farmer's income = rice yield × rice price – input costs;        (10)

$Nr_iA$ (kg N) represents the release amounts of certain Nr species (i), and $DC_i$ (¥ kg$^{-1}$ N) denotes
the damage cost (DC) per kg of certain Nr (i). $CO_2A$ (ton) and $DC_{CO2}$ (¥ ton$^{-1}$) represent the $CO_2$
emissions amount and global warming cost of $CO_2$, respectively. $N_2O$ is both a GHG and an Nr
species, but its environmental cost was calculated as a GHG here. The environmental costs mainly
refer to the global warming incurred by GHG emissions, soil acidification incurred by $NH_3$ and
$NO_X$ emissions, and aquatic eutrophication caused by $NH_3$ emissions, N leaching and runoff (Xia
and Yan, 2012).
**2.7 Nitrogen use efficiency**

Nitrogen use efficiency (NUE) is calculated by the following equation (Yan et al., 2014):

NUE = $(U_N-U_0)/F_N$;                        (11)

Here, $U_N$ is the plant N uptake (kg ha$^{-1}$) measured in aboveground biomass at physiological
maturity in the N fertilization treatments, while $U_0$ is the N uptake measured in aboveground



biomass in the treatment without N fertilizer addition (RN0). The N uptake in straw and grain was
analysed via concentrated sulfuric acid digestion and the Kjeldahl method (Zhao et al., 2015).

**2.8 Statistical analysis**

Differences in seasonal $CH_4$, $N_2O$ emissions and rice yield of the two rice-growing seasons
from 2013 to 2014 affected by fertilizer treatments, year and their interaction were examined by
using a two-way analysis of variance (ANOVA) (Table 2). The grain yield, seasonal $CH_4$ and
$N_2O$ emissions, SOCSR and GHGI of the different treatments were tested by analysis of variance
and mean values were compared by least significant difference (LSD) at the 5% level. All these
analyses were carried out using the SPSS (Version 19.0, USA).

**3 Results and discussion**

**3.1 Rice yield and NUE**

The two-way ANOVA analyses indicated that the rice grain yields were significantly affected
by the year and fertilizer treatment (Table 2). The farmer's practice plot (RN300) had an average
rice grain yield of 8395 kg ha$^{-1}$, with an NUE of 31.35%, over the two growing seasons from
2013 to 2014. Compared with RN300, reducing the N fertilizer rate by 20% (RN240) slightly
improved the grain yield and NUE to 8576 kg ha$^{-1}$ and 34.58%, respectively. Further N reduction,
without additional agricultural managements, could decrease the rice yield by 8.15% (RN180) and
15.18% (RN120) (Table 3). The response of rice yield to the synthetic N application rate in our
study successfully fitted a quadratic model (Fig.2), as has been reported in previous studies (Xia
and Yan, 2012; Cui et al., 2013a). Reducing N application to a reasonable rate, therefore, is
considered essential to reduce environmental costs, without sacrificing grain yield (Chen et al.,
2014). Lowering the N input adopted by local farmer (300 kg N ha$^{-1}$) by 20% could still enhance





the grain yield and NUE, without threatening food security in this study. However, a further
reduction of N 40% (RN180) would largely undermine the rice yield (Table 3).
Further reduction in N fertilizer may be achieved with improvements of agricultural
managements, Ju et al. (2009) reported that, based on knowledge-based N managements, such as
optimizing the N fertilizer source, rate, timing and place (in accordance with crop demand), rice
grain yield in the TLR was not significantly affected by a 30-60% N saving, while various Nr
losses would endure a two-fold curbing. Similarly, Zhao et al. (2015) found that the NUE could be
improved from 31% to 44%, even under a N reduction of 25% for rice production in the TLR,
through the implementation of integrated soil-crop system managements. In the present study, the
NUE was improved by 10% via a 20% N reduction, but it still falls behind the NUE in the studies
which received knowledge-based managements. Previous studies have proven that straw
incorporation exerted little positive impacts on grain yield. For instance, a meta-analysis
conducted by Singh et al. (2005) have found that incorporation of crop straw produced no
significant trend in improving crop yield in rice-based cropping systems. Moreover, based on a
long-term straw incorporation experiment established since 1990 at Changshu Agroecological
Experimental Station, Xia et al. (2014) have reported that long-term incorporation of wheat straw
only increased the rice yield by 1%. Therefore, in the present study, the effects of straw
incorporation on rice yield were considered as inappreciable.
**3.2 CH$_4$, N$_2$O emissions and SOSCR**
Over the two rice-growing seasons from 2013 to 2014, all treatments produced similar
patterns of CH$_4$ fluxes, albeit with large inter-annual variation (Fig.3a). The seasonal average CH$_4$
emissions from all plots showed no significant difference, ranging from 289.53 kg CH$_4$ ha$^{-1}$ in the





RN180 plot to 334.61 kg $CH_4$ $ha^{-1}$ in the RN120 plot (Table 4), much higher than observations
conducted in the same region (Zou et al., 2005; Ma et al., 2013). This phenomenon can be
attributed to the larger amounts of straw incorporation in this study (Table 1). Relative to the
RN300 plot, $CH_4$ emissions from the RN240 plot decreased by 8% and 10%, during the
rice-growing season of 2013 and 2014, respectively, although this effect was not statistically
significant (Table 4).

Many studies have shown a clear linear relationship between $CH_4$ emissions and the amounts

of applied organic matter (OM). Such an obvious linear relationship generally occurs under the
following conditions: first, the OM inputs are low (generally less than 3 Mg dry matter $ha^{-1}$) (Zou
et al., 2005; Ma et al., 2013); second, the applied OM rates among different treatments are
statistically different (Shang et al., 2011; Xia et al., 2014). It is possible that the linear response of
$CH_4$ emissions to OM inputs can become flat or even unobvious (Fig.S1), when OM is applied at
higher rates (in this study, the applied rates of straw in all N fertilization treatments were higher
than 4.4 Mg dry matter $ha^{-1}$) and these rates among treatments were not statistically different.
Besides, the experimental error caused by small differences in water conditions among different
treatments may also have promoted the unclear response of $CH_4$ emissions to straw inputs in this
study (Xia et al., 2014).

It is unsurprising that no obvious relationship between $CH_4$ emissions and N fertilizer

application rates was observed in this study (Fig.S1), because the effects of N fertilization on $CH_4$
production, transportation and oxidation are complex. For instance, N fertilization can provide
methanogens with more carbon substrates in the rhizosphere of plants by stimulating the growth of
rice biomass, thus promoting $CH_4$ production and transportation (Zou et al., 2005; Banger et al.,



2012). N enrichment could also enhance the activities of methanotrophs, therefore enhancing $CH_4$
oxidation (Xie et al., 2010; Yao et al., 2012). Moreover, ammonium-based fertilizer could compete
with $CH_4$ oxidation, due to the similar size and structure between $NH_4^+$ and $CH_4$ (Linquist et al.,
2012a).

The $N_2O$ fluxes were sporadic and pulse-like, and these fluxes showed large variations

between different seasons, and the majority of the $N_2O$ peaks occurred after the application of N
fertilizer (Fig.3b). The two-way ANOVA analyses indicated that the seasonal $N_2O$ emissions were
significantly affected by the year, the fertilizer treatment, and their interactions during the
rice-growing seasons (Table 2). The average $N_2O$ emission, during the two rice-growing seasons,
ranged from 0.05 kg N $ha^{-1}$ for the RN0 to 0.35 kg N $ha^{-1}$ for the RN300 (Table 4), which
increased exponentially as the N fertilizer rate increased. The average $N_2O$ emission factors varied
between 0.03% and 0.1%, with an average of 0.07%, which is comparable with previous studies
(0.05%-0.1%) conducted in the same region (Ma et al., 2013; Zhao et al., 2015).

The estimated topsoil (0-20cm) SOCSR varied from 0.130 t C $ha^{-1}$ $yr^{-1}$ for the RN0 plot to

0.197 t C $ha^{-1}$ $yr^{-1}$ for the RN300 plot (Table 4). The current SOCSR for rice production in the
TLR (0.197 t C $ha^{-1}$), falling within the SOCSR range of 0.13-2.20 t C$ha^{-1}$ $yr^{-1}$ estimated by Pan
et al. (2004) for paddy soils in China, is also comparable to the estimation of 0.17 t C $ha^{-1}$ $yr^{-1}$
from Ma et al. (2013) in a study based on a paddy field experiment in the same region. Moreover,
the provincial average SOCSR of Jiangsu province has been estimated to be 0.16-0.21 t C $ha^{-1}$ $yr^{-1}$
from the period of 1980 to 2000 (Huang & Sun, 2006, Liao et al., 2009), which is also similar to
our estimation.
**3.3 NGWP and GHGI**



The average NGWP for all treatments varied from 8656 to 11550 kg $CO_2$ eq $ha^{-1}$ (Table 4).
$CH_4$ emissions dominated the NGWP in all treatments, with the proportion ranging from 70.23%
to 88.56%, while synthetic N fertilizer production was the secondary contributor (Table 4). In
addition, SOC sequestration offset the positive GWP by 5.18-6.18% in the fertilization treatments.
Compared to conventional practice (RN300), the NGWP in the 20% reduction N practice (RN240)
decreased by 10.64%. Therein, 6.28% came from $CH_4$ reduction and 4.31% from N production
savings (Table 4). The GHGI of rice production ranged from 1.20 (RN240) to 1.61 (RN0) kg $CO_2$
eq $kg^{-1}$, which is higher than previous estimation of 0.24-0.74 kg $CO_2$ eq $kg^{-1}$ for rice production
in other rice-upland crop rotation systems (Qin et al., 2010; Ma et al., 2013). Moreover, the GHGI
of current rice production in the TLR (RW300) was estimated to be 1.45 times that of the national
average value estimated by Wang et al. (2014a), at 1.38 versus 0.95 kg $CO_2$ eq $kg^{-1}$.
Such phenomenon was attributed to the following reasons. First, compared to above studies,
current higher amounts of direct straw incorporation (2.9-6.2 Mg dry matter $ha^{-1}$), before rice
transplantation in the TLR, triggered substantial $CH_4$ emissions (290-335 kg $CH_4$ $ha^{-1}$). Crop
residue incorporation is regarded as a win-win strategy to benefit food security and mitigate
climate change, due to the fact that it possesses a large potential for carbon sequestration (Lu et al.,
2009). However, the GWP of straw-induced $CH_4$ emissions was reported to be 3.2-3.9 times that
of the straw-induced SOCSR, which indicates direct straw incorporation in paddy soils worsens
rather than mitigates climate changes, in terms of GWP (Xia et al., 2014). The SOC sequestration
induced by straw incorporation only offset the positive GWP by 5.2-6.2% in this study. Sensible
methods of straw incorporation should therefore be developed to reduce the substantial $CH_4$
emissions without compromising the build-up of SOC stock in the TLR. Second, the high N



application rate (300kg N ha$^{-1}$) in the TLR combined with the large emission factor of N fertilizer
manufacture, 8.3 kg $CO_2$-eq kg$^{-1}$ N (Zhang et al., 2013), promoted the sector of N fertilizer
production to be the secondary contributor to the GHGI (Table 4), while such sector wasn't
involved in above-mentioned studies. Compared to local farmer's practices (RN300), reducing the
N rate by 20% (RN240) lowered the GHGI by 13%, under the condition of straw incorporation,
although this effect was not statistically significant (Table 4). Compared to RN240, however,
further reduction of N rate (RN180 or RN120) increased the GHGI, largely due to the fact that rice
yield was considerably undermined under excessive N reduction. Therefore, the joint application
of reasonable N reduction and judicious method of straw incorporation would be promising in
reducing the GHGI for rice production in the TLR, in consideration of the current situation of
simultaneous high inputs of N fertilizer and wheat straw.
**3.4 Various Nr losses and NrI**

The results of the meta-analysis indicated that $N_2O$ emissions, as well as N leaching and

runoff, increase exponentially with an increase in N application rate (Fig.4b-d, $P < 0.01$), while
the response of $NH_3$ volatilization to N rates fitted the linear model best (Fig.4a, $P < 0.01$).
Established models can explain the variation in the estimation of various Nr losses by 50-57%.
The estimated total Nr losses for all treatments varied from 39.3 to 91.7 kg N ha$^{-1}$ in the
fertilization treatments (Table 5), accounting for 30.1-32.8% of N application rates. $NH_3$
volatilization dominated the NrI, with the proportion ranging from 53.5% to 57.4%, mainly
because of the current fertilizer application method (soil surface broadcasting) and high
temperatures in the field (Zhao et al., 2012b; Li et al., 2014). N runoff was the second most
important contributor, with the proportion ranging from 25.9% to 29.7% (Table 5). Using $^{15}$N





micro-plots combined with three-year field measurements, Zhao et al. (2012b) reported that the
total Nr loss from rice production in the TLR, under an N rate of 300 kg N ha$^{-1}$, was 98 kg N ha$^{-1}$,
which is comparable with our estimation of 91.69 kg N ha$^{-1}$ in the RN300 plot. Similarly, Xia and
Yan (2011) estimated the Nr loss for life-cycle rice production in this region to be around 90 kg N
ha$^{-1}$.

The NrI of rice production in different plots varied between 2.14 g N kg$^{-1}$ (RN0) and 10.92 g

N kg$^{-1}$ (RN300), which increased significantly as the N fertilizer rate increased (Table 5).The NrI
for rice production in the TLR was estimated to be 10.92 g N kg$^{-1}$ (RN300), which is 68% higher
than the national average value estimated by Chen et al. (2014), largely due to the higher N
fertilizer inputs in the TLR. Under the condition of straw incorporation, reducing the N application
rate by 20% pulled the NrI down to 8.42 g N kg$^{-1}$ (RN240) (Table 5). Additional N reduction
could further lower the NrI, but the rice yield would be compromised largely (Table 3). Previous
studies have proven that direct incorporation of crop straw exert unobvious effects on various Nr
releases (Xia et al., 2014). Because crop straws usually possess high values of C/N ratio and the
majority of N contented in the residue is not easily degraded by microorganisms in short-term period
(Huang et al., 2004). Therefore the straw incorporation could promote the N contained in the
residues to be stabilized in soil in long-term period, rather than directly releasing as various Nr
(Xia et al., 2014). For instance, a meta-analysis, integrating 112 scientific assessments of the crop
residue incorporation on the N$_2$O emissions, has reported that the practice exerted no statistically
significant effect on the N$_2$O releases (Shan and Yan, 2013). Therefore, the effects of wheat straw
incorporation on various Nr losses were considered as negligible in this study. Although no
specific relationship was found between the NrI and GHGI in all treatments in this study (Table 4





and Table 5), attention should be paid to the interrelationship between them. For instance, N
fertilizer production and application is an intermediate link between GHGI and NrI (Chen et al.,
2014). For the NrI, N fertilization promotes various Nr releases, exponentially or linearly (Fig.4),
while N production and application made a secondary contribution to the GHGI (Table 4). Such
interrelationships ought to be taken into account fully for any mitigation options pursued, in order
to reduce the GHG emissions and Nr discharges from rice production simultaneously (Cui et al.,
2013b; Cui et al., 2014).
**3.5 Economic evaluations of GHG emissions and Nr releases and their mitigation**
**potential**
The total environmental costs associated with the GHG emissions and Nr releases varied
from 1214 ¥ ha$^{-1}$ for the RN0 to 2399 ¥ ha$^{-1}$ for the RN300, which approximately accounted for
10.44-13.47% of the farmer's income and 27.05-32.47% of the input costs, respectively (Table 6).
$CH_4$ emission and $NH_3$ volatilization were the dominant contributors to the total environmental
costs, respectively (Table 4 and Fig.5). The total damage costs to environment accounted for 13.5%
of farmer's income under the current rice production in the TLR (RN300). Cutting the N rate from
300 to 240 kg N ha$^{-1}$ slightly improved the farmer's income by 3.64%, while further N reduction
would undermine the economic return of farmer's (Table 6).
GHG and Nr releases from rice production in the TLR are expected to possess a large
potential for mitigation, due to the current situation of direct straw incorporation and higher N
fertilizer inputs. Compared to traditional practice, a reduction of N application rate from 300 to
240 kg N ha$^{-1}$ could alleviate 12.52% for GHGI (Table 4), 22.94% for NrI (Table 5), and 15.76%
for environmental costs (Table 6). Further reduction in GHG and Nr releases (especially for $CH_4$



emissions and $NH_3$ volatilization) is possible, with the implementation of knowledge-based
managements (Chen et al., 2014; Nayak et al., 2015). For the mitigation of Nr releases, switching
the N fertilizer application method from surface broadcasting to deep incorporation could largely
lower the $NH_3$ volatilization from paddy soils (Zhang et al., 2012; Li et al., 2014). Moreover,
other optimum N managements, such as applying controlled-release fertilizers and nitrification or
urease inhibitors, could also effectively increase the NUE and reducing the overall Nr losses
(Chen et al., 2014). For the mitigation of GHG emissions, rather than being directly incorporated
before rice transplantation, crop residues should be preferentially decomposed under aerobic
conditions or used to produce biochar through pyrolysis, which could effectively reduce $CH_4$
emissions (Linquist et al., 2012b; Xie et al., 2013). Moreover, these pre-treatments are also
beneficial for carbon sequestration and food security (Woolf et al., 2010; Linquist et al., 2012b).

Most previous studies have merely focused on the quantification of GHG and Nr releases

from food production from the perspective of environment assessments (Zhao et al., 2012b; Ma et
al., 2013; Zhao et al., 2015). The perspective of economic evaluation is seldom implemented,
which goes against encouraging farmer to participate in the abatement of GHG and Nr releases on
their own initiative (Xia et al., 2014). The current pattern of rice production in the TLR incurs
great costs to the environment, which accounted for 13.47% of the net economic return that farmer
ultimately acquire (Table 6). Such an evaluation facilitates the translation of highly specialized
scientific conclusions into monetary-based information that is more familiar and accessible for
farmer, and therefore likely encouraging them to adopt eco-friendly agricultural managements
(Wang et al., 2014b). Profitability is generally considered the main driver for farmer to change
their management approach. Compared to traditional N application rate, a reduction of 20% would



make environmental costs savings of 14%, whilst simultaneously improving the economic return
of farmer's by 648 ¥ ha$^{-1}$ (Table 6). This represents an incentive for farmer to optimize their N
fertilizer application rates, provided that such information is available to them.
Considering the fact that no specific carbon- and Nr-mitigation incentive programs, like the
'Carbon Farming Initiative' in Australia (Lam et al., 2013), has been launched in China, an
ecological compensation incentive mechanism (national subsidy program) should be established
by governments. This would provide farmer with a tangible incentive, thus guiding them towards
gradually adopting knowledge-based managements, that could effectively curb GHG emissions
and Nr losses, but likely exert little positive effects on improving farmer's net economic return
(Xia et al., 2014). Examples include the composing of crop straws aerobically, or their use to
produce biochar before incorporation (Xie et al., 2013), and encouraging the deep placement of N
fertilizer (Wang et al., 2014b), as well as the application of enhanced-efficiency fertilizers during
the rice-growing season (Akiyama et al., 2010).

**4 Conclusions**

Our results demonstrated that producing per unit of rice yield released higher GHG and Nr in
the TLR, than that in other rice-upland cropping systems, which largely attributed to the current
situation of direct straw incorporation and excessive nitrogen fertilizer inputs. $CH_4$ emissions and
$NH_3$ volatilization dominated the GHG and Nr releases, respectively. Reducing the N application
rate by 20% from the tradition level (300 kg N ha$^{-1}$) could effectively decrease the GHG
emissions, Nr releases and the damage costs to the environment, while increased the rice yield and
improved farmer's income as well. Agricultural managements, such as making straw decompose
aerobically before incorporation and optimizing the application method of N fertilizer, could

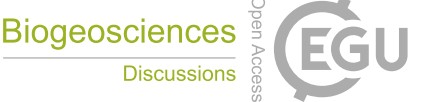

further reduce the GHG and Nr releases (especially $CH_4$ emissions and $NH_3$ volatilization) from
rice production in the TLR. Further studies are needed to evaluate the comprehensive effects of
these managements on GHG emissions, Nr releases and farmer's economic returns.

**Acknowledgements**

This study was financially supported by the CAS Strategic Priority Research Program (Grant
No. XDA05020200) and the National Science and Technology Pillar Program (2013BAD11B00).
We gratefully acknowledge the technical assistance provided by the Changshu Agroecological
Experimental Station of the Chinese Academy of Sciences.

**Supplementary material**

Supplementary material (Appendix A) associated with this article can be found, in the online
version.

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






**Table 1.** Field experimental treatments and agricultural management practices during
the rice-growing seasons of 2013 and 2014 in the TLR

| Treatment[a] | RN0 | RN120 | RN180 | RN240 | RN300 |
|---|---|---|---|---|---|
| Chemical fertilizer application rate ($N:P_2O_5:K_2O$, kg ha$^{-1}$) | 0:30:60 | 120:30:60 | 180:30:60 | 240:30:60 | 300:30:60 |
| Split N application ratio | --- | 4:3:3 | 4:3:3 | 4:3:3 | 4:3:3 |
| Straw application rate (Mg dry matter ha$^{-1}$) | 3.94/2.88[b] | 4.49/4.65 | 4.93/5.18 | 5.33/5.87 | 5.81/6.17 |
| Water regime[c] | F-D-F-M | F-D-F-M | F-D-F-M | F-D-F-M | F-D-F-M |
| Density ($10^4$ plants ha$^{-1}$) | 2.5 | 2.5 | 2.5 | 2.5 | 2.5 |

[a]RN0, RN120, RN180, RN240 and RN300 represent nitrogen application rates of 0, 120, 180, 240,
300 kg N ha$^{-1}$, respectively.
[b]3.94/2.88 denote that straw application rates during the rice-growing seasons of 2013 and 2014
are 3.94 and 2.88 Mg dry matter ha$^{-1}$, respectively.
[c]F, flooding; D, midseason drainage; M, moist but non-waterlogged by intermittent irrigation.








**Table 2.** Two-way ANOVA for the effects of fertilizer (F) application and year (Y) on $CH_4$ and
$N_2O$ emissions, and rice grain yields in rice paddies.

| Factor | df | $CH_4$ (kg ha$^{-1}$) | | | $N_2O$ (kg N ha$^{-1}$) | | | Yield (kg ha$^{-1}$) | | |
|---|---|---|---|---|---|---|---|---|---|---|
| | | SS | F | P | SS | F | P | SS | F | P |
| F | 4 | 8739 | 0.79 | 0.55 | 0.33 | 12.46 | $< 0.01$ | 39297547 | 32.96 | $< 0.01$ |
| Y | 1 | 4492 | 1.62 | 0.22 | 0.11 | 16.41 | $< 0.01$ | 2810414 | 9.43 | $< 0.01$ |
| F×Y | 4 | 2532 | 0.23 | 0.92 | 0.18 | 7.1 | $< 0.01$ | 750639 | 0.63 | 0.65 |
| Model | 9 | 15763 | 0.63 | 0.77 | 0.62 | 10.52 | $< 0.01$ | 42858600 | 15.97 | $< 0.01$ |
| Error | 16 | 20 | | | 0.13 | | | 5962260 | | |
















**Table 3.** Rice yield and NUE for the two rice-growing seasons from 2013 to 2014 in
the TLR

| Year | Treatment[a] | Yield (kg ha$^{-1}$) | NUE (%) |
|---|---|---|---|
| 2013 | RN0 | 4829 ± 207 | --- |
| | RN120 | 7079 ± 645 | 23.40 |
| | RN180 | 7655 ± 601 | 28.12 |
| | RN240 | 8273 ± 569 | 33.61 |
| | RN300 | 8029 ± 101 | 30.63 |
| 2014 | RN0 | 5919 ± 131 | --- |
| | RN120 | 7598 ± 1077 | 23.86 |
| | RN180 | 7768 ± 570 | 21.19 |
| | RN240 | 8880 ± 435 | 35.54 |
| | RN300 | 8761 ± 369 | 32.07 |
| Two-year average | RN0 | 5374 ± 617d[b] | --- |
| | RN120 | 7339 ± 843c | 23.63 |
| | RN180 | 7711 ± 527bc | 24.66 |
| | RN240 | 8576 ± 562a | 34.58 |
| | RN300 | 8395 ± 468ab | 31.35 |

[a]Definitions of the treatment codes are given in the footnotes of Table 1.
[b]Mean±SD; different letters within the same column indicate a significant difference at $p<0.05$.



**Table 4.** The NGWP and GHGI for the two rice-growing seasons from 2013 to 2014 in the TLR

| Year | Treatment[a] | $CH_4$ emission | $N_2O$ emission | SOCSR | Irrigation | N fertilizer production | Others | NGWP | GHGI |
|---|---|---|---|---|---|---|---|---|---|
| | | kg $CH_4$ ha$^{-1}$ | kg N ha$^{-1}$ | kg C ha$^{-1}$ yr$^{-1}$ | | kg $CO_2$ eq ha$^{-1}$ | | | kg $CO_2$ eq kg$^{-1}$ |
| 2013 | RN0 | 306.07 ± 41[b] | 0.08 ± 0.01 | 129.58 | 1170 | 0 | 217 | 8601 | 1.78 |
| | RN120 | 317.26 ± 92 | 0.10 ± 0.01 | 154.07 | 1170 | 996 | 265 | 9845 | 1.39 |
| | RN180 | 287.8 ± 12 | 0.13 ± 0.01 | 171.54 | 1170 | 1494 | 277 | 9568 | 1.25 |
| | RN240 | 273.27 ± 36 | 0.14 ± 0.06 | 185.50 | 1170 | 1992 | 291 | 9670 | 1.17 |
| | RN300 | 305.13 ± 90 | 0.16 ± 0.03 | 196.87 | 1170 | 2490 | 285 | 10927 | 1.36 |
| 2014 | RN0 | 307.22 ± 47 | 0.02 ± 0.05 | 129.58 | 1256 | 0 | 240 | 8711 | 1.47 |
| | RN120 | 351.96 ± 28 | 0.09 ± 0.02 | 154.07 | 1256 | 996 | 276 | 10805 | 1.42 |
| | RN180 | 291.25 ± 18 | 0.24 ± 0.04 | 171.54 | 1256 | 1494 | 280 | 9795 | 1.26 |



|  | RN240 | 317.65 ± 28 | 0.34 ± 0.12 | 185.50 | 1256 | 1992 | 303 | 10972 | 1.24 |
|---|---|---|---|---|---|---|---|---|---|
|  | RN300 | 343.8 ± 61 | 0.53 ± 0.21 | 196.87 | 1256 | 2490 | 301 | 12169 | 1.39 |
| Two-year | RN0 | 306.65 ± 39a | 0.05 ± 0.05b | 129.58c | 1213 | 0 | 229 | 8656 | 1.61 ± 0.25a |
| average | RN120 | 334.61 ± 64a | 0.09 ± 0.02b | 154.07bc | 1213 | 996 | 271 | 10322 | 1.40 ± 0.16b |
|  | RN180 | 289.53 ± 14a | 0.18 ± 0.07ab | 171.54ab | 1213 | 1494 | 279 | 9679 | 1.25 ± 0.09bc |
|  | RN240 | 295.46 ± 38a | 0.24 ± 0.14ab | 185.50ab | 1213 | 1992 | 297 | 10321 | 1.20 ± 0.08cd |
|  | RN300 | 324.47 ± 72a | 0.35 ± 0.25a | 196.87a | 1213 | 2490 | 293 | 11550 | 1.38 ± 0.21bc |

[a]Definitions of treatment codes are given in the footnotes of Table 1.

[b]Mean±SD; different letters within same column indicate a significant difference at $p < 0.05$.





**Table 5.** The seasonal average various Nr losses and NrI for the two rice-growing seasons from 2013 to 2014 in the TLR

| Treatment[a] | $NH_3$ volatilization | N runoff | N leaching | $N_2O$ emission | $NO_X$ emission | Total Nr losses | NrI |
|---|---|---|---|---|---|---|---|
| | | | kg N ha$^{-1}$ | | | | g N kg$^{-1}$ |
| RN0 | 0.64 | 5.39 | 1.44 | 0.07 | 3.96 | 11.50 | 2.14 |
| RN120 | 21.04 | 10.30 | 2.24 | 0.12 | 5.62 | 39.32 | 5.36 |
| RN180 | 31.24 | 14.25 | 2.80 | 0.21 | 6.44 | 54.93 | 7.12 |
| RN240 | 41.44 | 19.70 | 3.50 | 0.27 | 7.26 | 72.17 | 8.42 |
| RN300 | 51.64 | 27.24 | 4.37 | 0.38 | 8.07 | 91.69 | 10.92 |

[a]Definitions of treatment codes are given in the footnotes of Table 1.





**Table 6.** The seasonal average economic evaluation for rice production of the two growing seasons from 2013 to 2014 in the TLR (unit: ¥ ha$^{-1}$)

| Treatment[a] | Yield income[b] | Input costs[c] | Farmer's income[d] | Environmental costs[e] | |
| --- | --- | --- | --- | --- | --- |
| | | | | GHG emissions | Nr releases |
| RN0 | 16125 | 4493 | 11632 | 1143 | 71 |
| RN120 | 22020 | 6104 | 15916 | 1363 | 376 |
| RN180 | 23130 | 6542 | 16588 | 1278 | 535 |
| RN240 | 25725 | 7277 | 18448 | 1362 | 700 |
| RN300 | 25185 | 7385 | 17800 | 1525 | 874 |

[a]Definitions of treatment codes are given in the footnotes of Table 1.

[b]Yield income = rice yield × rice price.

[c]Input costs denote the economic input of purchasing various agricultural materials and hiring labours.

[d]Farmer's income = Yield income – input costs.

[e]Environmental costs denoted the sum of the acidification costs, eutrophication costs and global warming costs incurred by GHG emissions and Nr releases. The cost prices of GHG and Nr releases are as followed: GHG emission, 132 ¥ t$^{-1}$ CO$_2$ eq (Xia et al., 2014); NH$_3$ volatilization, 13.12 ¥ kg$^{-1}$ N; N leaching, 6.12 ¥ kg$^{-1}$ N; N runoff, 3.64 ¥ kg$^{-1}$ N; NO$_X$ emission, 8.7 ¥ kg$^{-1}$ N (Xia and Yan, 2011).

**Figure captions**



**Fig. 1. Seasonal variations in the daily precipitation and the temperature during the two rice–growing seasons of (a) 2013 and (b) 2014.**

**Fig.2. Relationship between N fertilizer application rate and seasonal average rice grain yield over the two rice-growing seasons of 2013 and 2014 in the TLR.** The vertical bars represent standard errors (n = 6).

**Fig.3. Seasonal variations in (a) CH$_4$ and (b) N$_2$O fluxes during the two rice-growing seasons from 2013 to 2014 in the TLR.** The arrow indicates N fertilizer application. The vertical bars represent standard errors (n = 3).

**Fig.4. Relationship between N fertilizer application rate and (a) NH$_3$ emissions, (b) N runoff, (c) N leaching and (d) N$_2$O emissions for rice production in the TLR.** These relationships were obtained through a meta-analysis.

**Fig.5. Seasonal average total environmental costs incurred by GHG emissions and Nr losses for rice production in TLR.**




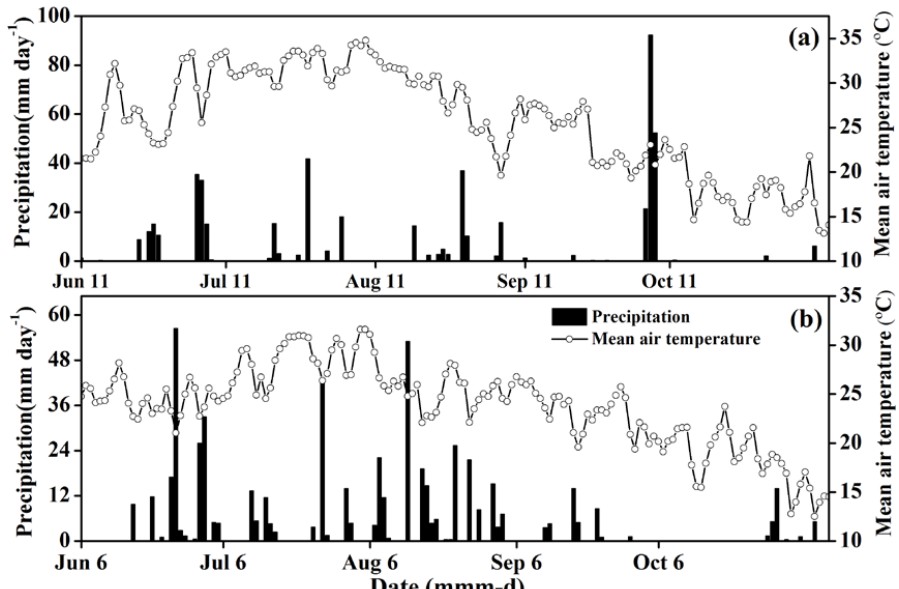

**Fig.1**





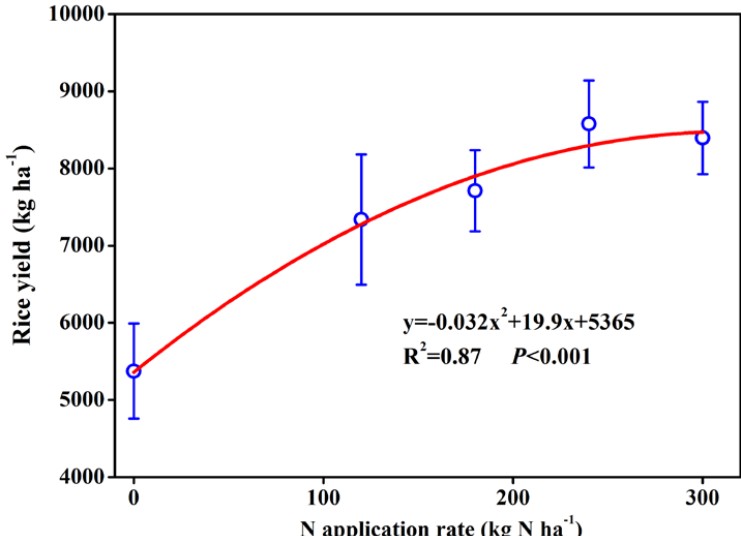

**Fig.2**




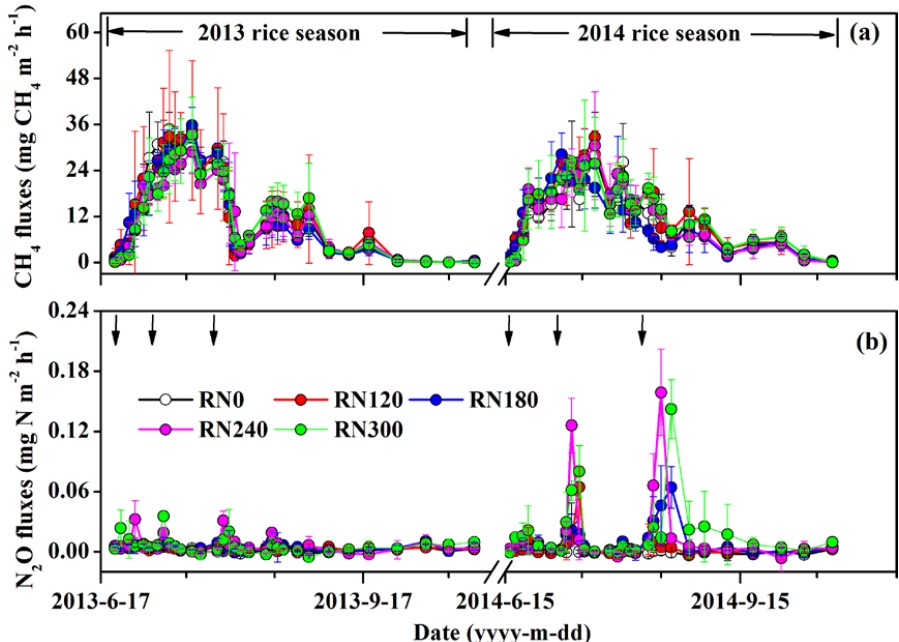

**Fig.3**





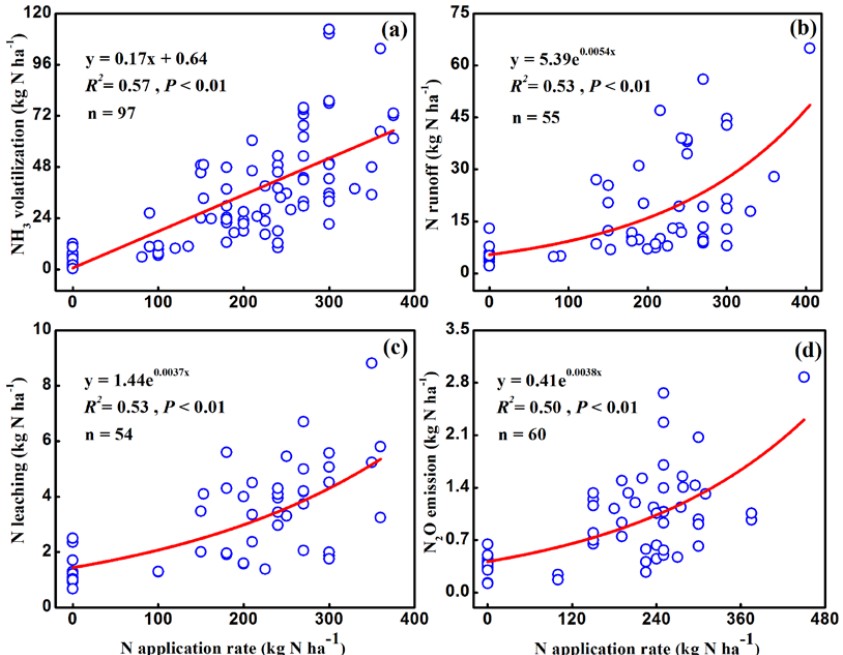

Fig.4



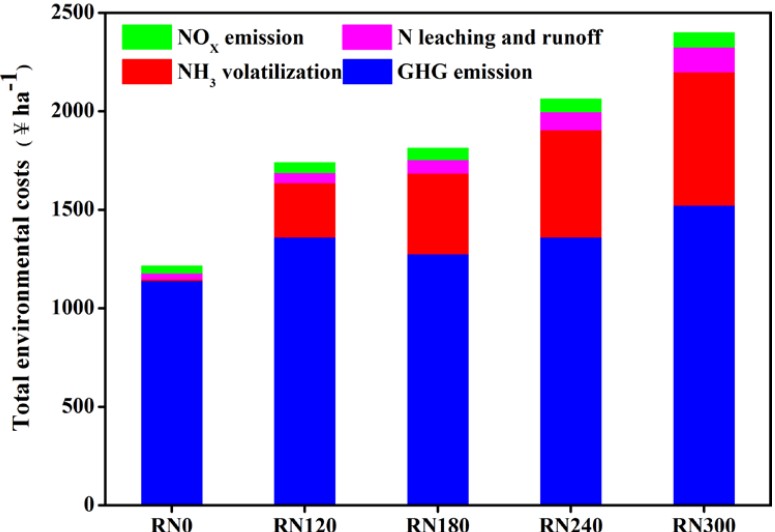

**Fig.5**