# Peer review of "Greenhouse gas emissions and reactive nitrogen releases from rice production with simultaneous incorporation of wheat straw and nitrogen fertilizer"

_Biogeosciences, 2015_

## Referee Comment (RC1) · Anonymous Referee #1 · 24 Feb 2016

Xia et al. evaluated the effects of N rates on GHG emission from rice paddy incorporated with wheat residues in Taihu Lake Region, and conducted an economic assessment on the environmental cost associated with agricultural practices. The manuscript is generally well written and its conclusion is supported by the data. A few comments:

The section on the details of the long-term experiment (lines 145-153) is not necessary. It was a bit confusing because the rates of residue incorporation used in this study (Tale 1) were different from those in the long-term experiment (line 146).

The relationship between CH4 emission and the amount of organic matter input of

different studies (lines 249-259) was not the major focus of the paper. The discussion should be simplified rather than being extended with possible explanations, some of which are speculative.

The section on the economic evaluation of the release and mitigation of GHGs and Nr was quite informative. At a few other places in the discussion section e.g. lines 278-285 the authors presented their results, and compared the results with others', which was fine but the manuscript would be more informative if the implications of the findings could be explored.

Minor comments:

Line 76: delete "And"

Line 112: The scientific name of rice was provided but not for winter wheat

Lines 252-253: What does it mean by "the applied OM rates among different treatments are statistically different"? A statistical test on the independent variables (OM application rates)??

Line 311: "was not" instead of "wasn't"

Lines 344-346: incomplete sentence

---

## Author Comment (AC1) · 10 Mar 2016

Dear Editors and Reviewer,

On behalf of my co-authors, thank you very much for your positive and constructive comments on our manuscript. We have carefully studied the comments and have made corrections which we hope to meet with approval. Please see the attached point-by-point responses and the tracked change version of manuscript for your further evaluation.

Response to Reviewer's comments: Reviewer 1: 1. The section on the details of the

long-term experiment (lines 145-153) is not necessary. Response: Thanks very much for your comment and suggestion. According to your suggestion, we have deleted some details of the long-term experiment and kept some to illustrate why we used equation (1) to calculate the SOCSR in this study (please see response to Question 2).

2. It was a bit confusing because the rates of residue incorporation used in this study (Table 1) were different from those in the long-term experiment (line 146). Response: Thanks for your comment and sorry for our unclear expression. Yes, the rates of residue incorporation used in this study were different from those in the long-term experiment, but we think it is appropriate to use the equation to calculate the SOCSR in this study, due to the following reasons. Generally, it takes long-term observations over years to decades before the SOC change is detectable (Yan et al., 2011). The SOC content changes of short-term field experiment couldn't be correctly measured, due to the high variability of SOC during the preliminary several years of the experiment. In this study, we used a relationship (based on the results of 22-year observation) between straw input rates and SOCSR, obtained via an on-going long-term straw application experiment in the same region, to calculate the SOCSR. Same agricultural management practices were applied to the on-going long-term experiment and the experiment of this study. Under the same agricultural managements, soil and climatic conditions, cropping systems and straw types, it is reasonable to believe that the rate of straw C stabilizing into SOC (i.e. conversion efficiency of crop residue C into SOC) are similar between these two experiments (Mandal et al., 2008). It is reported that the conversion rates of crop straw to SOC in two main wheat/maize production regions in China, which have similar soil and climatic conditions and agricultural practices, were very close, at 40.524 versus 40.607 kg SOC-C t$-1$ dry-weight straw (Lu et al., 2009). Therefore, we hold the opinion that the above SOCSR calculation method is appropriate, although the input rates of these two experiments were different. We have to admit that this method may bring uncertainty into our results, but it unlikely affects the main conclusions of our study (please see line 152-165). References: Yan, X., Cai, Z.,

[Figure]

Wang, S., Smith, P.: Direct measurement of soil organic carbon content change in the croplands of China, Global Change Biol., 17, 1487-1496, 2013. Mandal, B., Majumder, B., Adhya, T., Bandyopadhyay, P., Gangopadhyay, A., Sarkar, D., Kundu, M., Choudhury, S.G., Hazra, G., Kundu, S.: Potential of double-cropped rice ecology to conserve organic carbon under subtropical climate, Global Change Biol., 14, 2139-2151, 2008. Lu, F., Wang, X., Han, B., Ouyang, Z., Duan, X., Zheng, H., Miao, H.: Soil carbon sequestrations by nitrogen fertilizer application, straw return and no-tillage in China's cropland, Global Change Biol., 15, 281-305, 2009.

3. The relationship between CH4 emission and the amount of organic matter input was not the major focus of the paper. The discussion should be simplified rather than being extended with possible explanations, some of which are speculative. Response: Thanks for your good suggestion. According to your suggestion, we have simplified the relevant discussion (please see line 261-272).

4. At a few other places in the discussion section e.g. lines 278-285 the authors presented their results, and compared the results with others', which was fine but the manuscript would be more informative if the implications of the findings could be explored. Response: Thanks for your good suggestion. According to your suggestion, we have explored the implications of the SOC sequestration of this study (please see line 292-309). We also revised somewhere else, such as line 287-289 and line 359-360, to illustrate the implications of our findings.

5. Minor comments: Line 76: delete "And" Response: Agreed and revised (please see line 76). Line 112: The scientific name of rice was provided but not for winter wheat Response: Sorry for our carelessness. We have added the scientific name of wheat in the text (please see line 112).

Lines 252-253: What does it mean by "the applied OM rates among different treatments are statistically different"? A statistical test on the independent variables (OM application rates)? Response: Sorry for our unclear expression. We have deleted this

sentence.

Line 311: "was not" instead of "wasn't" Response: Agreed and revised (please see line 335).

Lines 344-346: incomplete sentence Response: Sorry for our unclear expression. We have revised this sentence (369-373).

Once again, thank you very much for your constructive comments and suggestions. Yours sincerely, XiaoyuanYan on behalf of all authors

Please also note the supplement to this comment:
http://www.biogeosciences-discuss.net/bg-2015-620/bg-2015-620-AC1-supplement.pdf

―――――――――――――――――

---

## Author Comment (AC2) · 10 Mar 2016

[revised manuscript text omitted]

Generally, it takes long-term observations over years to decades before the

SOC change is detectable (Yan et al.,

2011). The SOC content changes of  short-term field experiment couldn't be correctly measured , due to the high variability of SOC during the preliminary several years of the experiment. Therefore, we used the following relationship between the straw input rate (kg C $ha^{-1}$

$yr^{-1}$) and SOCSR (kg C $ha^{-1}$ $yr^{-1}$), obtained through an on-going long-term straw application experiment in the same region, to calculate the SOCSR in this study:

SOCSR = Straw input rate $\times$ 0.0603 + 31.39 ($R^2$ = 0.92);       (1)

This on-gonging long-term field experiment is also taking place at the Changshu

Agroecological Experimental Station (since 1990), which includes three straw application levels:

0, 4.5 t, and 9.0 t dry-weight $ha^{-1}$ $yr^{-1}$

The equation (1) was established based on the results of 22-year observation. Same agricultural management practices were applied to the  on-going long-term experiment and the experiment of this study . Under the same agricultural managements. , soil and climatic conditions, cropping systems and straw types, it is reasonable to believe that the rate of straw C stabilizing into SOC (i.e. conversion efficiency of crop residue C into SOC) are similar between these two experiments (Mandal et al., 2008). It is reported that the conversion rates of crop straw to SOC in two main wheat/maize production regions in China, which have similar climatic conditions and agricultural practices, were very close, at 40.524 versus 40.607 kg SOC-C $t^{-1}$ dry-weight straw (Lu et al., 2009). Therefore, we hold the opinion that the above SOCSR calculation method is appropriate, and the uncertainty incurred by this method unlikely affects the main conclusions of this study.

**2.4 Net global warming potential and greenhouse gas intensity**

The net global warming potential (NGWP, kg $CO_2$ eq $ha^{-1}$) and greenhouse gas intensity (GHGI, kg $CO_2$ eq $kg^{-1}$) of rice production in the TLR was calculated using the following equations:

$$NGWP = \sum_{i=1}^{m} AI_{ico_2} + CH_4 \times 25 + N_2O \times 44/28 \times 298 - SOCSR \times 44/12; \qquad (2)$$

$$GHGI = NGWP/rice\ yield; \qquad (3)$$

Here, $AI_{ico_2}$ denotes the GHG emissions from the production and transportation of agricultural inputs, which are calculated by multiplying their application rates by their individual GHG emission factors, such as synthetic fertilizers, diesel oil, electricity and pesticides (Liang, 2009; Zhang et al., 2013). $CH_4$ (kg $CH_4$ $ha^{-1}$), $N_2O$ (kg N $ha^{-1}$) and SOCSR (kg C $ha^{-1}$ $yr^{-1}$) represent the $CH_4$ emissions and $N_2O$ emissions from rice production, and the SOC sequestration rate, respectively.

**2.5 Total Nr losses and Nr intensity**

The total Nr losses (kg N ha$^{-1}$) and Nr intensity (NrI, g N kg$^{-1}$) were calculated using the following equations:

$\text{Total Nr losses} = \sum_{i=1}^{m} AI_{iN_r} + (NH_3 + N_2O + N_{Leaching} + N_{Runoff})_{rice}$;     (4)

$\text{NH}_3 \text{ volatilization} = 0.17 \times \text{N fertilizer rate} + 0.64$;     (5)

$\text{N runoff} = 5.39 \times \exp(0.0054 \times \text{N fertilizer rate})$;     (6)

$\text{N leaching} = 1.44 \times \exp(0.0037 \times \text{N fertilizer rate})$;     (7)

$\text{NrI} = (1000 \times \text{Total Nr losses}) / \text{rice yield}$;     (8)

[revised manuscript text omitted]
; this highlights that the reduction of N fertilizer rate is an effective approach to reduce the $N_2O$ emissions (Zou et al., 2005; Zhang et al., 2012). The average $N_2O$ emission factors varied between 0.03% and 0.1%, with an average of 0.07%, which is comparable with previous studies (0.05%-0.1%) conducted in the same region (Ma et al., 2013; Zhao et al., 2015).

The rice paddies have witnessed an increase in the SOC stock as a result of straw incorporation (Table 4). The estimated topsoil (0-20cm) SOCSR varied from 0.13 t C ha$^{-1}$ yr$^{-1}$ for the RN0 plot to 0.197 t C ha$^{-1}$ yr$^{-1}$ for the RN300 plot . The current SOCSR for rice production in the TLR (0.197 t C ha$^{-1}$), is comparable to the estimation of 0.17 t C ha$^{-1}$ yr$^{-1}$ from Ma et al. (2013) in a study based on a paddy field experiment with OM incorporation in the same region.  The magnitude of the SOC increase is variable depending on the straw incorporation method, the degree of tillage, the cropping systems and etc. (Yan et al., 2011; Huang et al., 2013). Liu et al. (2014) suggested that straw incorporation in rice-based cropping systems requires an overall consideration, due to the direct incorporation promoting substantial $CH_4$ emissions. When converting to $CO_2$-eq, the SOCSR only offsets the $CH_4$ emissions by 6.2-9.2% in this study (Table 4). This proportion is expected to increase provided that appropriate straw incorporation method (e.g., compost straw before incorporation) and conservative-tillage are adopted. The adoption of conservative-tillage system with straw return is proven to have advantages of increasing SOC stocks while reducing $CH_4$ emissions (Zhao et al.,

2015a; Zhao et al., 2015b).

**3.3 NGWP and GHGI**

[revised manuscript text omitted]

1068-1078, 2012b.

Zhao, X., Liu, S.L., Pu, C., Zhang, X.Q., Xue, J.F., Zhang, R., Wang, Y.Q., Lal, R., Zhang, H.L.,

Chen, F.:Methane and nitrous oxide emissions under no-till farming in China: a meta-analysis. Global Change Biol., 22, 1372-1384, 2015a.

Zhao, X., Zhang, R., Xue, J.F., Pu, C., Zhang, X.Q., Liu, S.L., Chen, F., Lal, R., Zhang, H.L.:

Management-induced changes to soil organic carbon in China: A meta-analysis. Adv.

Agron., 134, 1-49, 2015b.

Zou, J., Huang, Y., Jiang, J., Zheng, X., Sass, R.L.: A 3-year field measurement of methane and nitrous oxide emissions from rice paddies in China: Effects of water regime, crop residue, and fertilizer application, Global Biogeochem. Cycles, 19, 2005.

**Table 1.** Field experimental treatments and agricultural management practices during the rice-growing seasons of 2013 and 2014 in the TLR

| Treatment[a] | RN0 | RN120 | RN180 | RN240 | RN300 |
|---|---|---|---|---|---|
| Chemical fertilizer application rate (N:$P_2O_5$:$K_2O$, kg ha$^{-1}$) | 0:30:60 | 120:30:60 | 180:30:60 | 240:30:60 | 300:30:60 |
| Split N application ratio | --- | 4:3:3 | 4:3:3 | 4:3:3 | 4:3:3 |
| Straw application rate (Mg dry matter ha$^{-1}$) | 3.94/2.88[b] | 4.49/4.65 | 4.93/5.18 | 5.33/5.87 | 5.81/6.17 |
| Water regime[c] | F-D-F-M | F-D-F-M | F-D-F-M | F-D-F-M | F-D-F-M |
| Density ($10^4$ plants ha$^{-1}$) | 2.5 | 2.5 | 2.5 | 2.5 | 2.5 |

[a]RN0, RN120, RN180, RN240 and RN300 represent nitrogen application rates of 0, 120, 180, 240,

300 kg N ha$^{-1}$, respectively.

[b]3.94/2.88 denote that straw application rates during the rice-growing seasons of 2013 and 2014

are 3.94 and 2.88 Mg dry matter ha$^{-1}$, respectively.

[c]F, flooding; D, midseason drainage; M, moist but non-waterlogged by intermittent irrigation.

**Table 2.** Two-way ANOVA for the effects of fertilizer (F) application and year (Y) on $CH_4$ and

$N_2O$ emissions, and rice grain yields in rice paddies.

| Factor | df | $CH_4$ (kg ha$^{-1}$) | | | $N_2O$ (kg N ha$^{-1}$) | | | Yield (kg ha$^{-1}$) | | |
|---|---|---|---|---|---|---|---|---|---|---|
| | | SS | F | P | SS | F | P | SS | F | P |
| F | 4 | 8739 | 0.79 | 0.55 | 0.33 | 12.46 | < 0.01 | 39297547 | 32.96 | < 0.01 |
| Y | 1 | 4492 | 1.62 | 0.22 | 0.11 | 16.41 | < 0.01 | 2810414 | 9.43 | < 0.01 |
| F×Y | 4 | 2532 | 0.23 | 0.92 | 0.18 | 7.1 | < 0.01 | 750639 | 0.63 | 0.65 |
| Model | 9 | 15763 | 0.63 | 0.77 | 0.62 | 10.52 | < 0.01 | 42858600 | 15.97 | < 0.01 |
| Error | 16 | 20 | | | 0.13 | | | 5962260 | | |

**Table 3.** Rice yield and NUE for the two rice-growing seasons from 2013 to 2014 in the TLR

| Year | Treatment[a] | Yield (kg ha$^{-1}$) | NUE (%) |
|------|------|------|------|
| 2013 | RN0 | $4829 \pm 207$ | --- |
| | RN120 | $7079 \pm 645$ | 23.40 |
| | RN180 | $7655 \pm 601$ | 28.12 |
| | RN240 | $8273 \pm 569$ | 33.61 |
| | RN300 | $8029 \pm 101$ | 30.63 |
| 2014 | RN0 | $5919 \pm 131$ | --- |
| | RN120 | $7598 \pm 1077$ | 23.86 |
| | RN180 | $7768 \pm 570$ | 21.19 |
| | RN240 | $8880 \pm 435$ | 35.54 |
| | RN300 | $8761 \pm 369$ | 32.07 |

| Two-year average | RN0 | $5374 \pm 617d^{b}$ | --- |
| | RN120 | $7339 \pm 843c$ | 23.63 |
| | RN180 | $7711 \pm 527bc$ | 24.66 |
| | RN240 | $8576 \pm 562a$ | 34.58 |
| | RN300 | $8395 \pm 468ab$ | 31.35 |

[a]Definitions of the treatment codes are given in the footnotes of Table 1.

[b]Mean±SD; different letters within the same column indicate a significant difference at $p<0.05$.

**Table 4.** The NGWP and GHGI for the two rice-growing seasons from 2013 to 2014 in the TLR

| Year | Treatment[a] | $CH_4$ emission | $N_2O$ emission | SOCSR | Irrigation | N fertilizer production | Others | NGWP | GHGI |
|------|-----------|-------------|-------------|-------|------------|-------------------------|--------|------|------|
| | | kg $CH_4$ ha$^{-1}$ | kg N ha$^{-1}$ | kg C ha$^{-1}$ yr$^{-1}$ | kg $CO_2$ eq ha$^{-1}$ | | | | kg $CO_2$ eq kg$^{-1}$ |
| 2013 | RN0 | $306.07 \pm 41$[b] | $0.08 \pm 0.01$ | 129.58 | 1170 | 0 | 217 | 8601 | 1.78 |
| | RN120 | $317.26 \pm 92$ | $0.10 \pm 0.01$ | 154.07 | 1170 | 996 | 265 | 9845 | 1.39 |
| | RN180 | $287.8 \pm 12$ | $0.13 \pm 0.01$ | 171.54 | 1170 | 1494 | 277 | 9568 | 1.25 |
| | RN240 | $273.27 \pm 36$ | $0.14 \pm 0.06$ | 185.50 | 1170 | 1992 | 291 | 9670 | 1.17 |
| | RN300 | $305.13 \pm 90$ | $0.16 \pm 0.03$ | 196.87 | 1170 | 2490 | 285 | 10927 | 1.36 |
| 2014 | RN0 | $307.22 \pm 47$ | $0.02 \pm 0.05$ | 129.58 | 1256 | 0 | 240 | 8711 | 1.47 |
| | RN120 | $351.96 \pm 28$ | $0.09 \pm 0.02$ | 154.07 | 1256 | 996 | 276 | 10805 | 1.42 |
| | RN180 | $291.25 \pm 18$ | $0.24 \pm 0.04$ | 171.54 | 1256 | 1494 | 280 | 9795 | 1.26 |

| | | | | | | | | | |
|---|---|---|---|---|---|---|---|---|---|
| | RN240 | 317.65 ± 28 | 0.34 ± 0.12 | 185.50 | 1256 | 1992 | 303 | 10972 | 1.24 |
| | RN300 | 343.8 ± 61 | 0.53 ± 0.21 | 196.87 | 1256 | 2490 | 301 | 12169 | 1.39 |
| Two-year average | RN0 | 306.65 ± 39a | 0.05 ± 0.05b | 129.58c | 1213 | 0 | 229 | 8656 | 1.61 ± 0.25a |
| | RN120 | 334.61± 64a | 0.09 ± 0.02b | 154.07bc | 1213 | 996 | 271 | 10322 | 1.40 ± 0.16b |
| | RN180 | 289.53 ± 14a | 0.18 ± 0.07ab | 171.54ab | 1213 | 1494 | 279 | 9679 | 1.25 ± 0.09bc |
| | RN240 | 295.46 ± 38a | 0.24 ± 0.14ab | 185.50ab | 1213 | 1992 | 297 | 10321 | 1.20 ± 0.08cd |
| | RN300 | 324.47 ± 72a | 0.35 ± 0.25a | 196.87a | 1213 | 2490 | 293 | 11550 | 1.38 ± 0.21bc |

[a]Definitions of treatment codes are given in the footnotes of Table 1.

[b]Mean±SD; different letters within same column indicate a significant difference at $p<0.05$.

**Table 5.** The seasonal average various Nr losses and NrI for the two rice-growing seasons from 2013 to 2014 in the TLR

| Treatment[a] | $NH_3$ volatilization | N runoff | N leaching | $N_2O$ emission | $NO_X$ emission | Total Nr losses | NrI |
|---|---|---|---|---|---|---|---|
| | kg N ha$^{-1}$ | | | | | | g N kg$^{-1}$ |
| RN0 | 0.64 | 5.39 | 1.44 | 0.07 | 3.96 | 11.50 | 2.14 |
| RN120 | 21.04 | 10.30 | 2.24 | 0.12 | 5.62 | 39.32 | 5.36 |
| RN180 | 31.24 | 14.25 | 2.80 | 0.21 | 6.44 | 54.93 | 7.12 |
| RN240 | 41.44 | 19.70 | 3.50 | 0.27 | 7.26 | 72.17 | 8.42 |
| RN300 | 51.64 | 27.24 | 4.37 | 0.38 | 8.07 | 91.69 | 10.92 |

[a]Definitions of treatment codes are given in the footnotes of Table 1.

**Table 6.** The seasonal average economic evaluation for rice production of the two growing seasons from 2013 to 2014 in the TLR (unit: ¥ ha$^{-1}$)

| Treatment[a] | Yield income[b] | Input costs[c] | Farmer's income[d] | Environmental costs[e] | |
|---|---|---|---|---|---|
| | | | | GHG emissions | Nr releases |
| RN0 | 16125 | 4493 | 11632 | 1143 | 71 |
| RN120 | 22020 | 6104 | 15916 | 1363 | 376 |
| RN180 | 23130 | 6542 | 16588 | 1278 | 535 |
| RN240 | 25725 | 7277 | 18448 | 1362 | 700 |
| RN300 | 25185 | 7385 | 17800 | 1525 | 874 |

[a]Definitions of treatment codes are given in the footnotes of Table 1.

[b]Yield income = rice yield × rice price.

[c]Input costs denote the economic input of purchasing various agricultural materials and hiring labours.

[d]Farmer's income = Yield income – input costs.

[e]Environmental costs denoted the sum of the acidification costs, eutrophication costs and global warming costs incurred by GHG emissions and Nr releases. The cost prices of GHG and Nr releases are as followed: GHG emission, 132 ¥ t$^{-1}$ $CO_2$ eq (Xia et al., 2014); $NH_3$ volatilization, 13.12 ¥ kg$^{-1}$ N; N leaching, 6.12 ¥ kg$^{-1}$ N; N runoff, 3.64 ¥ kg$^{-1}$ N; $NO_X$ emission, 8.7 ¥ kg$^{-1}$ N (Xia and Yan, 2011).

**Figure captions**

**Fig. 1. Seasonal variations in the daily precipitation and the temperature during the two rice–growing seasons of (a) 2013 and (b) 2014.**

**Fig.2. Relationship between N fertilizer application rate and seasonal average rice grain yield over the two rice-growing seasons of 2013 and 2014 in the TLR.** The vertical bars represent standard errors (n = 6).

**Fig.3. Seasonal variations in (a) $CH_4$ and (b) $N_2O$ fluxes during the two rice-growing seasons from 2013 to 2014 in the TLR.** The arrow indicates N fertilizer application. The vertical bars represent standard errors (n = 3).

**Fig.4. Relationship between N fertilizer application rate and (a) $NH_3$ emissions, (b) N runoff, (c) N leaching and (d) $N_2O$ emissions for rice production in the TLR.** These relationships were obtained through a meta-analysis.

**Fig.5. Seasonal average total environmental costs incurred by GHG emissions and Nr losses for rice production in TLR.**

[Figure]

**Fig.1**

[Figure]

**Fig.2**

[Figure]

**Fig.3**

[Figure]

**Fig.4**

[Figure]

**Fig.5**

---

## Referee Comment (RC2) · Anonymous Referee #2 · 26 May 2016

Xia et al. investigated GHG emission and Nr losses from rice production in response to applications of N fertilizer and wheat straw. They looked into the total environmental costs incurred by the GHG emission and Nr losses. Such study is important for the comprehensive evaluation of impacts of GHG and Nr losses on environment. The methods used in this study are appropriate, and the results are well discussed. I recommend it for publication in 'Biogeosciences', if the following questions are well considered.

Specific comments: Abstract: Authors employed the meta-analysis to calculate the

various Nr losses. As an important part of this study, the results of the meta-analysis should be simply presented in the abstract. Moreover, it would be better if the abstract is concisely shortened, since some findings in the current version were insignificant, e.g., L34 'while methane emission . . ...wheat rates increased'.

L71-72. Specify the current water and straw application methods.

L140. Using the relationship of straw input rate and SOCSR of previous study to calculate the SOC changes in this study is fine, since both of the studies have similar climatic conditions, cropping history and agricultural practices. But the uncertainty should be noticed and can be discussed in the result and discussion part.

L193-205. The environmental cost evaluation is interesting. But, why treated N2O as a GHG when conduced this evaluation, since it is both a GHG and Nr species?

L275-280. This discussion needs to be concise, since the effect of N fertilizer on CH4 emission is beyond the focus of this study.

L289-290. The calculation of the N2O emission factor needs to be specified in the methodology.

L345. Does the straw application affect the Nr losses (e.g., N2O and NH3 emission) and the subsequent calculation of Nr intensity?

L377-378. I don't think the GHGI and Nr have to have some specific relationship, although the N production and fertilization can both affect them.

L428. The 'ecological compensation mechanism' is a good idea to encourage famers to adopt knowledge-based agricultural managements. To make it clearer, authors need to provide more details about that rather than just giving a mention.

Some further remarks Although the main text is generally well written, some grammar errors should be corrected carefully. L 72, delete 'the' L 98-101, long sentence, needs to be split. L102, N2O should be 'nitrous oxide (N2O) L116, delete 'an' L196, 'was'

should be 'were' L230, replace 'to a reasonable rate' with 'reasonably' L233, delete 'without threatening food...study' L252, replace 'produced' with 'showed' L335, 'manufacture' should be 'production' L348, delete the sentence L427, 'has' should be 'have' L443, delete 'as well' Tables 1-6, the abbreviations in the table titles should be self-explained.

---

## Author Comment (AC3) · 1 Jun 2016

Dear Prof. Richard Conant and Reviewer, On behalf of my co-authors, thank you very much for your positive and constructive comments on our manuscript. We have carefully studied the comments and have made corrections which we hope to meet with approval. Please see the attached point-by-point responses and the tracked change version of manuscript for your further evaluation. All revised positions mentioned in the responses can be readily found in the attached clear version of manuscript.

Response to Reviewer's comments: Reviewer 2: Specific comments 1. 1. Abstract:

Authors employed the meta-analysis to calculate the various Nr losses. As an important part of this study, the results of the meta-analysis should be simply presented in the abstract. Moreover, it would be better if the abstract is concisely shortened, since some findings in the current version were insignificant, e.g., L34 'while methane emission . ....wheat rates increased'. Response: Thanks very much for your comment and suggestion. According to your suggestion, we have presented the main findings of the meta-analysis in the abstract. We have also concisely shortened the abstract (please see Line 24-53).

2. L71-72, specify the current water and straw application methods. Response: Thanks for your comment and sorry for our unclear expression. We have specified the water and straw application methods (please see Line 78-79).

3. L140 Using the relationship of straw input rate and SOCSR of previous study to calculate the SOC changes is fine, since both of the studies have similar climatic conditions, cropping history and agricultural practices. But the uncertainty should be noticed and can be discussed in the result and discussion part. Response: Thanks for your good suggestion. According to your suggestion, we have noticed the uncertainty induced by the SOCSR calculation method and discussed it in the results and discussion part of 'CH4, N2O emissions and SOSCR'. Moreover, we also presented the reasons why we hold the opinion that the SOCSR calculation method in this study is appropriate, and the uncertainty incurred by this method unlikely affects the main conclusions of this study (please see Line 305-323).

4. L193-205. The environmental cost evaluation is interesting. But, why treated N2O as a GHG when conduced this evaluation, since it is both a GHG and Nr species? Response: Thanks for your comment. N2O is both a GHG and Nr species, but its environmental cost was calculated as a GHG here. This is because the cost of N2O emission as Nr species is mainly to damage human health (Gu et al., 2012). But the effects of Nr losses on the direct damage costs of human health were not included in this study, which are very difficult to quantify. The environmental costs included in

this study mainly refer to the global warming incurred by GHG emissions, soil acidification incurred by NH3 and NOX emissions, and aquatic eutrophication caused by NH3 emissions, N leaching and runoff (Xia and Yan, 2012). We have added such reasons in the methodology to make it clearer (please see Line 207-209). References: Gu, B., Ge, Y., Ren, Y., Xu, B., Luo, W., Jiang, H., Gu, B., Chang, J.: Atmospheric reactive nitrogen in China: Sources, recent trends, and damage costs, Environ. Sci. Technol., 46, 9420-9427, 2012. Xia, Y., Yan, X.: Ecologically optimal nitrogen application rates for rice cropping in the Taihu Lake region of China, Sustain. Sci., 7, 33-44, 2012.

5. L275-280. This discussion needs to be concise, since the effect of N fertilizer on CH4 emission is beyond the focus of this study. Response: Thanks for your suggestion. According to your suggestion, we have simplified the relevant discussion (please see Line 291-293).

6. L289-290. The calculation of the N2O emission factor needs to be specified in the methodology. Response: Thanks for your correction. According to your suggestion, we have specified the calculation of the N2O emission factor in the methodology (please see Line 217-222).

7. L345. Does the straw application affect the Nr losses (e.g., N2O and NH3 emission) and the subsequent calculation of Nr intensity? Response: Thanks for your comment. Previous studies have proven that direct incorporation of crop straw had insignificant effects on various Nr releases (Xia et al., 2014). Because the majority of N contented in the crop straw is not easily degraded by microorganisms in a short-term period, and can be stabilized in soil in a long-term period, rather than being released as various Nr (Huang et al., 2004; Xia et al., 2014). For instance, a meta-analysis, integrating 112 scientific assessments of the crop residue incorporation on the N2O emissions, has reported that the practice exerted no statistically significant effect on the N2O releases (Shan and Yan, 2013). Therefore, the effects of wheat straw incorporation on various Nr losses were considered as negligible in this study. Moreover, previous studies have also proven that straw incorporation exerted little impacts on grain yield. For instance,

a meta-analysis conducted by Singh et al. (2005) have found that incorporation of crop straw produced no significant trend in improving crop yield in rice-based cropping systems. Moreover, based on a long-term straw incorporation experiment established since 1990 in the TLR, Xia et al. (2014) have reported that long-term incorporation of wheat straw only increased the rice yield by 1%. Therefore, in the present study, the effects of straw incorporation on NrI were considered as inappreciable. We have presented such reasons in the results and discussion part to make it clearer (please see Line 255-262 and Line 396-405). References: Huang, Y., Zou, J., Zheng, X., Wang, Y., Xu, X.: Nitrous oxide emissions as influenced by amendment of plant residues with different C: N ratios, Soil Biol. Biochem., 36, 973-981, 2004. Shan, J., Yan, X.Y.: Effects of crop residue returning on nitrous oxide emissions in agricultural soils, Atmos. Environ., 71, 170-175, 2013. Singh, Y., Singh, B., Timsina, J.: Crop residue management for nutrient cycling and improving soil productivity in rice-based cropping systems in the tropics, Adv. Agron., 85, 269-407, 2005. Xia, L., Wang, S., Yan, X.: Effects of long-term straw incorporation on the net global warming potential and the net economic benefit in a rice-wheat cropping system in China, Agric. Ecosyst. Environ., 197, 118-127, 2014.

8. L377-378. I don't think the GHGI and Nr have to have some specific relationship, although the N production and fertilization can both affect them. Response: Thanks for your comment and sorry for our unclear expression. We have deleted such sentence. What we wanted to present is that extra attention should be paid to the interrelationship between the NrI and GHGI, which could provide hints for the mitigation purpose. For instance, N fertilizer production and application is an intermediate link between the NrI and GHGI (Chen et al., 2014). For the NrI, N fertilization promotes various Nr releases, exponentially or linearly (Fig.4), while N production and application made a secondary contribution to the GHGI (Table 4). Such interrelationships ought to be taken into account fully for any mitigation options pursued, in order to reduce the GHG emissions and Nr discharges from rice production simultaneously (Cui et al., 2013b; Cui et al., 2014) (please see Line 408-415). References: Chen, X., Cui, Z., Fan, M., Vitousek,

P., Zhao, M., Ma, W., Wang, Z., Zhang, W., Yan, X., Yang, J.: Producing more grain with lower environmental costs, Nature, 514, 486-489, 2014. Cui, Z., Yue, S., Wang, G., Zhang, F., Chen, X.: In-season root-zone N management for mitigating greenhouse gas emission and reactive N losses in intensive wheat production, Environ. Sci. Technol., 47, 6015-6022, 2013b. Cui, Z., Wang, G., Yue, S., Wu, L., Zhang, W., Zhang, F., Chen, X.: Closing the N-use efficiency gap to achieve food and environmental security, Environ. Sci. Technol., 48, 5780-5787, 2014.

9. L428. The 'ecological compensation mechanism' is a good idea to encourage famers to adopt knowledge-based agricultural managements. To make it clearer, authors need to provide more details about that rather than just giving a mention. Response: Thanks for your good suggestion. According to your suggestion, we have added more details to make the 'ecological compensation mechanism' clearer (please see Line 458-467).

Reviewer 2: Some further remarks 1. L 72, delete 'the' Response: Thanks for your correction. We have revised it according to your correction (please see Line 80).

2. L 98-101, long sentence, needs to be split. Response: Thanks for your correction. We have revised it according to your correction (please see Line 105-108).

3. L102, N2O should be 'nitrous oxide (N2O) Response: Thanks for your correction. We have revised it according to your correction (please see Line 110).

4. L116, delete 'an' Response: Thanks for your correction. We have revised it according to your correction (please see Line 124).

5. L196, 'was' should be 'were' Response: Thanks for your correction. We have revised it according to your correction (please see Line 201).

6. L230, replace 'to a reasonable rate' with 'reasonably' Response: Thanks for your correction. We have revised it according to your correction (please see Line 241).

7. L233, delete 'without threatening food…study' Response: Thanks for your correction. We have revised it according to your correction (please see Line 244-245).

8. L252, replace 'produced' with 'showed' Response: Thanks for your correction. We have revised it according to your correction (please see Line 264).

9. L335, 'manufacture' should be 'production' Response: Thanks for your correction. We have revised it according to your correction (please see Line 360).

10. L348, delete the sentence Response: Thanks for your correction. We have revised it according to your correction (please see Line 375).

11. L427, 'has' should be 'have' Response: Thanks for your correction. We have revised it according to your correction (please see Line 459).

12. L443, delete 'as well' Response: Thanks for your correction. We have revised it according to your correction (please see Line 478).

13. Table 1-6, the abbreviations in the table titles should be self-explained. Response: Thanks for your correction. We have revised it according to your correction (please see the tables).

Once again, thank you very much for your constructive comments and suggestions.

In addition, we also polished the English expressions in the whole manuscript and redrew Figure 5. All changes in the manuscript will not influence the main conclusions of the paper. And here we did not list the changes but marked in red in the attached tracked change version of manuscript. We appreciate Editor/Reviewer's warm work earnestly, and hope that the correction will meet with approval.

Yours sincerely, XiaoyuanYan on behalf of all authors

Please also note the supplement to this comment:
http://www.biogeosciences-discuss.net/bg-2015-620/bg-2015-620-AC3-supplement.pdf

---

## Author Comment (AC4) · 1 Jun 2016

**Greenhouse gas emissions and reactive nitrogen releases from rice production with simultaneous incorporation of wheat straw and nitrogen fertilizer**

Longlong Xia[a,b], Yongqiu Xia[a], Shutan Ma[a,b], Jinyang Wang[a], Shuwei Wang[a,b], Wei Zhou[a,b], Xiaoyuan Yan[a*]

[a.] State Key Laboratory of Soil and Sustainable Agriculture, Institute of Soil Science, Chinese Academy of Sciences, Nanjing 210008, China.

[b.] University of Chinese Academy of Sciences, Beijing 100049, China.

[*]**Corresponding author**: Xiaoyuan Yan

State Key Laboratory of Soil and Sustainable Agriculture, Institute of Soil Science, Chinese Academy of Sciences, Nanjing 210008, P. R. China

Phone number: +86 025 86881530, Fax: +86 025 86881000

Email address: yanxy@issas.ac.cn

**Abstract**

Impacts of simultaneous inputs of crop straw and nitrogen (N) fertilizer on greenhouse gas (GHG) emissions and N losses from rice production  are not well understood. A two-year field experiment was established in a rice-wheat cropping system in the Taihu Lake region (TLR) of China  to evaluate the GHG intensity (GHGI), reactive N intensity (NrI)  of rice production with inputs of wheat straw and N fertilizer . The field experiment included five treatments of different N fertilization rates for rice production: 0 (RN0), 120 (RN120), 180 (RN180), 240 (RN240) and 300 kg N ha$^{-1}$ (RN300, traditional N applied rate in the TLR). Wheat straws were fully incorporated into soil before rice transplantation  The meta-analytic technique was employed to evaluate various Nr losses. Results showed that the response of rice yield to N  rate successfully fitted a quadratic model, while N fertilization promoted Nr discharges exponentially (nitrous oxide emission, N leaching and runoff) or linearly (ammonia volatilization). The GHGI of rice production ranged from 1.20 (RN240) to 1.61 (RN0) kg $CO_2$-equivalent ($CO_2$-eq) kg$^{-1}$ . while NrI varied from 2.14 (RN0) to 10.92 (RN300) g N kg$^{-1}$ . Methane ($CH_4$ ) emission dominated GHGI with proportion of 70.2-88.6% due to direct straw incorporation, while ammonia ($NH_3$) volatilization dominated NrI with proportion of 53.5-57.4%. Damage costs to environment incurred by GHG and Nr releases from current rice production (RN300) accounted for 8.8% and 4.9% of farmers' incomes, respectively. Cutting  N application rate  from 300 (traditional N rate)

to 240 kg N ha$^{-1}$ could improve rice yield and nitrogen use efficiency by 2.14% and

10.30%, respectively, whilst simultaneously reduced GHGI by 13%, NrI by 23% and total environmental costs by 16%. Moreover, the reduction of 60 kg N ha$^{-1}$ improved farmers'

income by 639 ¥ha$^{-1}$, which would provide them with an incentive to change  the current N application rate. Our study suggests that GHG and Nr releases, especially  CH$_4$

emission and NH$_3$ volatilization, from rice production in the TLR could be further reduced, considering the current incorporation pattern of wheat straw and N fertilizer.

[revised manuscript text omitted]

Generally, it takes long-term observations over years to decades before the soil organic carbon (SOC) change is detectable (Yan et al.,

2011). The SOC content changes of  short-term field experiment couldn't be correctly measured , due to the high variability of SOC during the preliminary several years of the experiment. Therefore, we used the following relationship between the straw input rate (kg C ha$^{-1}$

yr$^{-1}$) and SOC sequestration rate (SOCSR, kg C ha$^{-1}$ yr$^{-1}$), obtained through an on-going long-term straw application experiment in the same region, to calculate the SOCSR in this study

$$SOCSR = \text{Straw input rate} \times 0.0603 + 31.39 \ (R^2 = 0.92); \qquad (1)$$

This on-gonging long-term field experiment is also taking place at the Changshu Agroecological Experimental Station (since 1990), which includes three straw application levels: 0, 4.5 t, and 9.0 t dry-weight $ha^{-1}$ $yr^{-1}$  The equation (1) was established based on the results of 22-year observation (Xia et al., 2014). Same agricultural management practices were applied to the on-going long-term experiment and the experiment of this study.

**2.4 Net global warming potential and greenhouse gas intensity**

The net global warming potential (NGWP, kg $CO_2$ eq $ha^{-1}$) and greenhouse gas intensity (GHGI, kg $CO_2$ eq $kg^{-1}$) of rice production in the TLR was calculated using the following equations:

$$NGWP = \sum_{i=1}^{m} AI_{ico_2} + CH_4 \times 25 + N_2O \times 44/28 \times 298 - SOCSR \times 44/12; \qquad (2)$$

$$GHGI = NGWP/\text{rice yield}; \qquad (3)$$

Here, $AI_{ico_2}$ denotes the GHG emissions from the production and transportation of agricultural inputs, which are calculated by multiplying their application rates by their individual GHG emission factors, such as synthetic fertilizers, diesel oil, electricity and pesticides (Liang, 2009; Zhang et al., 2013). $CH_4$ (kg $CH_4$ $ha^{-1}$), $N_2O$ (kg N $ha^{-1}$) and SOCSR (kg C $ha^{-1}$ $yr^{-1}$) represent the CH₄  and N₂O emissions from rice production, and the SOC sequestration rate, respectively.

**2.5 Total Nr losses and Nr intensity**

The total Nr losses (kg N ha$^{-1}$) and Nr intensity (NrI, g N kg$^{-1}$) were calculated using the following equations:

$$\text{Total Nr losses} = \sum_{i=1}^{m} AI_{iN_r} + (NH_3 + N_2O + N_{Leaching} + N_{Runoff})_{rice}; \qquad (4)$$

$$NH_3 \text{ volatilization} = 0.17\times   \times N_{rate} + 0.64;$$

(5)

$$N \text{ runoff} = 5.39 \times Exp~(0.0054    \times N_{rate});$$

(6)

$$N \text{ leaching} = 1.44 \times Exp~(0.0037\times   ~~rate);N_{rate});$$

(7)

$$NrI = (1000 \times \text{Total Nr losses}) / \text{rice yield}; \qquad (8)$$

Here, $AI_{i_{Nr}}$ denotes the Nr lost (mainly through N₂O and NO$_X$ emissions) from the production and transportation of agricultural inputs (Liang, 2009; Zhang et al., 2013), while

'(NH₃+N₂O+N$_{Leaching}$+N$_{Runoff}$)$_{rice}$' represents the NH₃ volatilization, N₂O emissions, N leaching and runoff during the rice-growing season.  N$_{rate}$ represents the N fertilizer application rate. Nr empirical models (Equation 5, 6, 7) derived from a meta-analysis of published literature  concerning Nr losses

from rice production in the TLR. Specific details regarding this literature survey are provided in

Appendix A.

**2.6 Total environmental costs incurred by GHG and Nr releases and farmer's income**

The total environmental costs (¥ $ha^{-1}$) incurred by GHG and Nr releases and farmer's income from rice production in the TLR were calculated based on the following equations:

$$\text{Environmental costs} = \sum_{i=1}^{n}(Nr_iA \times DC_i) + CO_2A \times DC_{CO2}; \qquad (9)$$

$$\text{Farmer's income} = \text{rice yield} \times \text{rice price} - \text{input costs}; \qquad (10)$$

$Nr_iA$ (kg N) represents the release amounts of certain Nr species (i), and $DC_i$ (¥ $kg^{-1}$ N) denotes the damage cost (DC) per kg of certain Nr (i). $CO_2A$ (ton) and $DC_{CO2}$ (¥ $ton^{-1}$) represent the $CO_2$ emissions amount and global warming cost of $CO_2$, respectively. $N_2O$ is both a GHG and  Nr species, but its environmental cost was calculated as a GHG here. Because the cost of $N_2O$ emission as Nr species is to damage human health (Gu et al., 2012), but the effects of Nr losses on the damage costs of human health were not included in this study. The environmental costs mainly refer to the global warming incurred by GHG emissions, soil acidification incurred by $NH_3$ and $NO_X$ emissions, and aquatic eutrophication caused by $NH_3$ emissions, N leaching and runoff (Xia and Yan, 2012).

**2.7 Nitrogen use efficiency and $N_2O$ emission factor**

Nitrogen use efficiency (NUE) and $N_2O$ emission factor ($EF_d\%$) were respectively calculated by the following  (equations (Ma et al., 2013; Yan et al., 2014):

$$NUE = (U_N - U_0)/F_N; \qquad (11)$$

$$EF_d\% = (E_N - E_0)/F_N; \qquad (12)$$

Here, $U_N$ is the plant N uptake (kg $ha^{-1}$) measured in aboveground biomass at physiological maturity in the N fertilization treatments, while $U_0$ is the N uptake measured in aboveground biomass in the treatment without N fertilizer addition (RN0). $E_N$ denotes the cumulative $N_2O$

emissions in the N fertilization treatments, while $E_0$ denotes the $N_2O$ emissions in the RN0. $F_N$

[revised manuscript text omitted]

The rice paddies have witnessed an increase in the SOC stock as a result of straw incorporation (Table 4). The estimated topsoil (0—20cm) SOCSR varied from 0.13013 t C $ha^{-1}$

$yr^{-1}$ for the RN0 plot to 0.197 t C $ha^{-1}$ $yr^{-1}$ for the RN300 plot (Table 4). The . The empirical model established through a long-term straw incorporation study in the same region was employed to evaluate the SOCSR in this study, which likely brought uncertainty into the results of this study. Under the same agricultural managements, soil and climatic conditions, cropping systems and straw types, it is reasonable to believe that the rates of straw C stabilizing into SOC (i.e. conversion efficiency of crop residue C into SOC) are similar between these two experiments (Mandal et al., 2008). It is reported that the conversion rates of crop straw to SOC in two main wheat/maize production regions in China, which have similar climatic conditions and agricultural practices, were very close, at 40.524 versus 40.607 kg SOC-C $t^{-1}$ dry-weight straw (Lu et al., 2009). Moreover, the current estimated SOCSR for rice production in the TLR (0.197 t C $ha^{-1}$), is comparable to the estimation of 0.17 t C $ha^{-1}$ $yr^{-1}$ from Ma et al. (2013) in a study based on a paddy field experiment with OM incorporation in the same region. Therefore, we hold the opinion that the above SOCSR calculation method is appropriate, and the uncertainty incurred by this method unlikely affects the main conclusions of this study.

The magnitude of the SOC increase is variable depending on the straw incorporation method, the degree of tillage, the cropping systems and etc. (Yan et al., 2011; Huang  et al., 2013). Liu et al. (2014) suggested that straw incorporation in rice-based cropping systems requires an overall consideration, due to the direct incorporation promoting substantial $CH_4$ emissions. When converting to $CO_2$ eq, the SOCSR only offsets the $CH_4$ emissions by 6.2–9.2% in this study (Table 4). This proportion is expected to increase provided that appropriate straw incorporation method (e.g., compost straw before incorporation) and conservative-tillage are adopted. Moreover, previous studies have shown that the combined adoption of conservative-tillage system with straw return had large advantages in increasing SOC stocks while reducing $CH_4$ emissions (Zhao et al., 2015a; Zhao et al., 2015b).

**3.3 NGWP and GHGI**

The average NGWP for all treatments varied from 8656 to 11550 kg $CO_2$ eq $ha^{-1}$ (Table 4). $CH_4$ emissions dominated the NGWP in all treatments, with the proportion ranging from 70.23% to 88.56%, while synthetic N fertilizer production was the secondary contributor (Table 4). In addition, SOC sequestration offset the positive GWP by 5.18–6.18% in the fertilization treatments. Compared to conventional practice (RN300), the NGWP in the 20% reduction N practice (RN240) decreased by 10.64%. Therein, 6.28% came from $CH_4$ reduction and 4.31% from N production savings (Table 4). The GHGI of rice production ranged from 1.20 (RN240) to 1.61 (RN0) kg $CO_2$ eq $kg^{-1}$, which is higher than previous estimation of 0.24–0.74 kg $CO_2$ eq $kg^{-1}$ for rice production in other rice-upland crop rotation systems (Qin et al., 2010; Ma et al., 2013). Moreover, the GHGI of current rice production in the TLR (RW300) was estimated to be 1.45 times that of the national average value estimated by Wang et al. (2014a), at 1.38 versus 0.95 kg $CO_2$ eq $kg^{-1}$.

Such phenomenon was attributed to the following reasons. First, compared to above studies, current higher amounts of direct straw incorporation (2.9–6.2 Mg dry matter $ha^{-1}$), before rice transplantation in the TLR, triggered substantial $CH_4$ emissions (290–335 kg $CH_4$ $ha^{-1}$). Crop residue incorporation is regarded as a win-win strategy to benefit food security and mitigate climate change, due to the fact that it possesses a large potential for carbon sequestration (Lu et al.,

2009). However, the GWP of straw-induced $CH_4$ emissions was reported to be 3.2–3.9 times that of the straw-induced SOCSR, which indicates direct straw incorporation in paddy soils worsens rather than mitigates climate changes, in terms of GWP (Xia et al., 2014). The SOC sequestration induced by straw incorporation only offset the positive GWP by 5.2–6.2% in this study. Sensible methods of straw incorporation should therefore be developed to reduce the substantial $CH_4$

emissions without compromising the build-up of SOC stock in the TLR.

Second, the high N application rate (300kg N $ha^{-1}$) in the TLR combined with the large emission factor of N fertilizer production, 8.3 kg $CO_2$-eq $kg^{-1}$ N (Zhang et al., 2013), marked the sector of N fertilizer production as the secondary contributor to the

GHGI (Table 4); this sector , however, was not involved in above-mentioned studies. Compared to local farmer's practices (RN300), reducing the N rate by 20% (RN240)

lowered the GHGI by 13%, under the condition of straw incorporation, although this effect was not statistically significant (Table 4). Compared to RN240, however, further reduction of N rate (RN180 or RN120) increased the GHGI,  due to the fact that rice yield was considerably reduced under excessive N reduction. Therefore, the joint application of reasonable N

reduction and judicious method of straw incorporation would be promising in reducing the GHGI

for rice production in the TLR, in consideration of the current situation of simultaneous high inputs of N fertilizer and wheat straw.

**3.4 Various Nr losses and NrI**

The results of the meta-analysis indicated that $N_2O$ emissions, as well as N leaching and runoff, increased exponentially with an increase in N application rate (Fig.4b-d, $P < 0.01$), while the response of $NH_3$ volatilization to N rates fitted the linear model best (Fig.4a, $P < 0.01$).

 The estimated total Nr losses for all treatments varied from 39.3 to 91.7 kg N ha$^{-1}$ in the fertilization treatments (Table 5), accounting for 30.1–32.8% of N application rates. NH$_3$ volatilization dominated the NrI, with the proportion ranging from 53.5% to 57.4%, mainly because of the current fertilizer application method (soil surface broadcast) and high temperatures in the field (Zhao et al., 2012b; Li et al., 2014). N runoff was the second most important contributor (Table 5). Using $^{15}$N micro-plots combined with three-year field measurements, Zhao et al. (2012b) reported that the total Nr losses from rice production in the TLR, under an N rate of 300 kg N ha$^{-1}$, were 98 kg N ha$^{-1}$, which is comparable with our estimation of 91.69 kg N ha$^{-1}$ in the RN300 plot. Similarly, Xia and Yan (2011) estimated the Nr losses for life-cycle rice production in this region to be around 90 kg N ha$^{-1}$. The high proportion (30.1–32.8%) of the applied N fertilizer released as Nr from rice production in the TLR, highlights the need to adopt reasonable N managements to increase the plant N uptake and reduce Nr losses (Ju et al., 2009).

The NrI of rice production in different plots varied between 2.14 g N kg$^{-1}$ (RN0) and 10.92 g N kg$^{-1}$ (RN300), which increased significantly as the N fertilizer rate increased (Table 5). The NrI for rice production in the TLR was estimated to be 10.92 g N kg$^{-1}$ (RN300), which is 68% higher than the national average value estimated by Chen et al. (2014), as a result of higher N fertilizer input in the TLR. Under the condition of straw incorporation, reducing N application rate by 20% pulled the NrI down to 8.42 g N kg$^{-1}$ (RN240) (Table 5). Additional N reduction could further lower the NrI, but the rice yield would be compromised largely (Table 3). Previous studies have proven that direct incorporation of crop straw had insignificant effects on various Nr releases (Xia et al., 2014). Because  the majority of N contented in the  crop straw is not easily degraded by microorganisms in a short-term period  and can be stabilized in soil in a long-term period, rather than being released as various Nr (Huang et al., 2004; Xia et al., 2014). For instance, a meta-analysis, integrating 112 scientific assessments of the crop residue incorporation on the $N_2O$ emissions, has reported that the practice exerted no statistically significant effect on the $N_2O$ releases (Shan and Yan, 2013). Therefore, the effects of wheat straw incorporation on various Nr losses were considered as negligible in this study.

Extra attention should be paid to the interrelationship between the NrI and GHGI, which could provide hints for the mitigation purpose. For instance, N fertilizer production and application is an intermediate link between the NrI and GHGI  (Chen et al., 2014). For the NrI, N fertilization promotes various Nr releases, exponentially or linearly (Fig.4), while N production and application made a secondary contribution to the GHGI (Table 4). Such interrelationships ought to be taken into account fully for any mitigation options pursued, in order to reduce the GHG emissions and Nr discharges from rice production simultaneously (Cui et al., 2013b; Cui et al., 2014).

**3.5 Economic evaluations of GHG emissions and Nr releases and their mitigation potential**

The total environmental costs associated with the GHG emissions and Nr releases varied from 1214 ¥ha$^{-1}$ for the RN0 to 2399 ¥ha$^{-1}$ for the RN300, which approximately accounted for

10.44–13.47% of the farmer's income and 27.05–32.47% of the input costs, respectively (Table

6). CH$_4$ emission and NH$_3$ volatilization were the dominant contributors to the total environmental costs, respectively (Table 4 and Fig.5). The total damage costs to environment accounted for 13.5%

of farmer's income under the current rice production in the TLR (RN300). Cutting the N rate from

300 to 240 kg N ha$^{-1}$ slightly improved the farmer's income by 3.64%, while further N reduction would reduce the economic return of farmer's (Table 6).

GHG and Nr releases from rice production in the TLR are expected to possess a large potential for mitigation, due to the current situation of direct straw incorporation and higher N

fertilizer inputs. Compared to traditional practice, a reduction of N application rate from 300 to

240 kg N ha$^{-1}$ could alleviate 12.52% for GHGI (Table 4), 22.94% for NrI (Table 5), and 15.76%

for environmental costs (Table 6). Further reduction in GHG and Nr releases (especially for CH$_4$

emissions and NH$_3$ volatilization) is possible, with the implementation of knowledge-based managements (Chen et al., 2014; Nayak et al., 2015). For the mitigation of Nr releases, switching the N fertilizer application method from surface broadcast to deep incorporation could largely lower the NH$_3$ volatilization from paddy soils (Zhang et al., 2012; Li et al., 2014).

Moreover, other optimum N managements, such as applying controlled-release fertilizers and urease inhibitors, could also effectively increase the NUE and reducing the overall

Nr losses (Chen et al., 2014). For the mitigation of GHG emissions, rather than being directly incorporated before rice transplantation, crop residues should be preferentially decomposed under aerobic conditions or used to produce biochar through pyrolysis, which could effectively reduce

CH$_4$ emissions (Linquist et al., 2012; Xie et al., 2013). Moreover, these pre-treatments are also beneficial for carbon sequestration and yield production (Woolf et al., 2010;

Linquist et al., 2012).

Most previous studies have merely focused on the quantification of GHG and Nr releases from food production from the perspective of environment assessments (Zhao et al., 2012b; Ma et al., 2013; Zhao et al., 2015). The perspective of economic evaluation is seldom implemented, which goes against encouraging farmer to participate in the abatement of GHG and Nr releases on their own initiative (Xia et al., 2014). The current pattern of rice production in the TLR incurs great costs to the environment, which accounted for 13.47% of the net economic return that farmer ultimately acquire (Table 6). Such an evaluation facilitates the translation of highly specialized scientific conclusions into monetary-based information that is more familiar and accessible for farmers, and therefore likely encouraging them to adopt eco-friendly agricultural managements (Wang et al., 2014b). Profitability is generally considered the main driver for farmer to change their management approach. Compared to traditional N application rate, a reduction of

20% would make environmental costs savings of 14%, whilst simultaneously improving the economic return of farmer's by 648 ¥ ha$^{-1}$ (Table 6). This represents an incentive for farmers to optimize their N fertilizer application rates, provided that such information is available to them.

Considering the fact that no specific carbon- and Nr-mitigation incentive programs, like the

'Carbon Farming Initiative' in Australia (Lam et al., 2013), have been launched in China, an ecological compensation incentive mechanism  should be established by governments.  This should be a national subsidy program with a special compensation and award fund to cover the extra mitigation costs induced by the adoption of knowledge-based mitigation managements for farmers (Xia et al., 2016). Such a program would provide farmer with a tangible incentive, thus guiding them towards gradually adopting the mitigation managements, which could effectively curb GHG emissions and Nr losses, but likely exert little positive effects on improving their net economic return (Xia et al., 2014). Examples include the composing of crop straws aerobically, or their use to produce biochar before incorporation (Xie et al., 2013), and encouraging the application of deep placement of N fertilizer (Wang et al., 2014b), as well as the application of enhanced-efficiency N fertilizers during the rice-growing season (Akiyama et al., 2010).

**4 Conclusions**

Our results demonstrated that producing  rice yield in the TLR released substantial GHG and Nr , which largely attributed to the current direct straw incorporation and excessive N fertilizer inputs. $CH_4$ emissions and $NH_3$ volatilization dominated the GHG and Nr releases, respectively. Reducing  N application rate by 20% from the tradition level (300 kg N ha$^{-1}$) could effectively decrease the GHG emissions, Nr releases and the damage costs to the environment, while increased the rice yield and improved farmer's income simultaneously. Agricultural managements, such as making straw decompose aerobically before its incorporation and optimizing the application method of N fertilizer, showed large potentials to further reduce the GHG (e.g., $CH_4$ emission) and Nr releases (e.g., $NH_3$ volatilization) from rice production in this region. Further studies are needed to evaluate the comprehensive effects of these managements on GHG emissions, Nr releases and farmer's economic returns.

**Acknowledgements**

[revised manuscript text omitted]

1068-1078, 2012b.

Zhao, X., Liu, S.L., Pu, C., Zhang, X.Q., Xue, J.F., Zhang, R., Wang, Y.Q., Lal, R., Zhang, H.L.,

Chen, F.:Methane and nitrous oxide emissions under no-till farming in China: a meta-analysis, Global Change Biol., 22, 1372-1384, 2015a.

Zhao, X., Zhang, R., Xue, J.F., Pu, C., Zhang, X.Q., Liu, S.L., Chen, F., Lal, R., Zhang, H.L.:

Management-induced changes to soil organic carbon in China: A meta-analysis. Adv.

Agron., 134, 1-49, 2015b.

Zou, J., Huang, Y., Jiang, J., Zheng, X., Sass, R.L.: A 3-year field measurement of methane and nitrous oxide emissions from rice paddies in China: Effects of water regime, crop residue, and fertilizer application, Global Biogeochem. Cycles, 19, 2005.

**Table 1.** Field experimental treatments and agricultural management practices during the rice-growing seasons of 2013 and 2014 in the TLRTaihu Lake region

| Treatment[a] | RN0 | RN120 | RN180 | RN240 | RN300 |
|---|---|---|---|---|---|
| Chemical fertilizer | 0:30:60 | 120:30:60 | 180:30:60 | 240:30:60 | 300:30:60 |

| | | | | | |
|---|---|---|---|---|---|
| application rate | | | | | |
| $(N:P_2O_5:K_2O, kg\ ha^{-1})$ | | | | | |
| Split N application ratio | --- | 4:3:3 | 4:3:3 | 4:3:3 | 4:3:3 |
| Straw application rate $(Mg\ dry\ matter\ ha^{-1})$ | $3.94/2.88^b$ | 4.49/4.65 | 4.93/5.18 | 5.33/5.87 | 5.81/6.17 |
| Water regime[c] | F-D-F-M | F-D-F-M | F-D-F-M | F-D-F-M | F-D-F-M |
| Density $(10^4\ plants\ ha^{-1})$ | 2.5 | 2.5 | 2.5 | 2.5 | 2.5 |

[a]RN0, RN120, RN180, RN240 and RN300 represent N application rates of 0, 120, 180,

240, 300 kg N ha$^{-1}$, respectively.

[b]3.94/2.88 denote that straw application rates during the rice-growing seasons of 2013 and 2014

are 3.94 and 2.88 Mg dry matter ha$^{-1}$, respectively.

[c]F, flooding; D, midseason drainage; M, moist but non-waterlogged by intermittent irrigation.

**Table 2.** Two-way ANOVA for the effects of fertilizer (F) application and year (Y) on CH$_4$ and

N$_2$O emissions, and rice grain yields in rice paddies.

| Factor | df | CH$_4$ (kg ha$^{-1}$) | | | N$_2$O (kg N ha$^{-1}$) | | | Yield (kg ha$^{-1}$) | | |
|---|---|---|---|---|---|---|---|---|---|---|
| | | SS | F | P | SS | F | P | SS | F | P |

| | | | | | | | | | | |
|---|---|---|---|---|---|---|---|---|---|---|
| F | 4 | 8739 | 0.79 | 0.55 | 0.33 | 12.46 | < 0.01 | 39297547 | 32.96 | < 0.01 |
| Y | 1 | 4492 | 1.62 | 0.22 | 0.11 | 16.41 | < 0.01 | 2810414 | 9.43 | < 0.01 |
| F×Y | 4 | 2532 | 0.23 | 0.92 | 0.18 | 7.1 | < 0.01 | 750639 | 0.63 | 0.65 |
| Model | 9 | 15763 | 0.63 | 0.77 | 0.62 | 10.52 | < 0.01 | 42858600 | 15.97 | < 0.01 |
| Error | 16 | 20 | | | 0.13 | | | 5962260 | | |

**Table 3.** Rice yield and nitrogen use efficiency (NUE) for the two rice-growing seasons from 2013

to 2014 in the Taihu Lake region

| Year | Treatment[a] | Yield (kg ha$^{-1}$) | NUE (%) |
|---|---|---|---|
| 2013 | RN0 | 4829 ± 207 | --- |
| | RN120 | 7079 ± 645 | 23.40 |

| | | | |
|---|---|---|---|
| | RN180 | 7655 ±601 | 28.12 |
| | RN240 | 8273 ±569 | 33.61 |
| | RN300 | 8029 ±101 | 30.63 |
| 2014 | RN0 | 5919 ±131 | --- |
| | RN120 | 7598 ±1077 | 23.86 |
| | RN180 | 7768 ±570 | 21.19 |
| | RN240 | 8880 ±435 | 35.54 |
| | RN300 | 8761 ±369 | 32.07 |
| Two-year average | RN0 | 5374 ±617d[b] | --- |
| | RN120 | 7339 ±843c | 23.63 |
| | RN180 | 7711 ±527bc | 24.66 |
| | RN240 | 8576 ±562a | 34.58 |
| | RN300 | 8395 ±468ab | 31.35 |

[a]Definitions of the treatment codes are given in the footnotes of Table 1.

[b]Mean±SD; different letters within the same column indicate a significant difference at $p < 0.05$.

**Table 4.** The net global warming potential (NGWP) and greenhouse gas intensity (GHGI) for the two rice-growing seasons from 2013 to 2014 in the Taihu Lake region

| Year | Treatment[a] | CH$_4$ emission | N$_2$O emission | SOCSR | Irrigation | N fertilizer production | Others | NGWP | GHGI |
|------|--------------|-----------------|-----------------|-------|------------|-------------------------|--------|------|------|
| | | kg CH$_4$ ha$^{-1}$ | kg N ha$^{-1}$ | kg C ha$^{-1}$ yr$^{-1}$ | kg CO$_2$ eq ha$^{-1}$ | | | | kg CO$_2$ eq kg$^{-1}$ |
| 2013 | RN0 | 306.07 ±41[b] | 0.08 ±0.01 | 129.58 | 1170 | 0 | 217 | 8601 | 1.78 |
| | RN120 | 317.26 ±92 | 0.10 ±0.01 | 154.07 | 1170 | 996 | 265 | 9845 | 1.39 |
| | RN180 | 287.8 ±12 | 0.13 ±0.01 | 171.54 | 1170 | 1494 | 277 | 9568 | 1.25 |
| | RN240 | 273.27 ±36 | 0.14 ±0.06 | 185.50 | 1170 | 1992 | 291 | 9670 | 1.17 |
| | RN300 | 305.13 ±90 | 0.16 ±0.03 | 196.87 | 1170 | 2490 | 285 | 10927 | 1.36 |
| 2014 | RN0 | 307.22 ±47 | 0.02 ±0.05 | 129.58 | 1256 | 0 | 240 | 8711 | 1.47 |
| | RN120 | 351.96 ±28 | 0.09 ±0.02 | 154.07 | 1256 | 996 | 276 | 10805 | 1.42 |

| | | | | | | | | |
|---|---|---|---|---|---|---|---|---|
| | RN180 | 291.25 ±18 | 0.24 ±0.04 | 171.54 | 1256 | 1494 | 280 | 9795 | 1.26 |
| | RN240 | 317.65 ±28 | 0.34 ±0.12 | 185.50 | 1256 | 1992 | 303 | 10972 | 1.24 |
| | RN300 | 343.8 ±61 | 0.53 ±0.21 | 196.87 | 1256 | 2490 | 301 | 12169 | 1.39 |
| Two-year average | RN0 | 306.65 ±39a | 0.05 ±0.05b | 129.58c | 1213 | 0 | 229 | 8656 | 1.61 ±0.25a |
| | RN120 | 334.61 ±64a | 0.09 ±0.02b | 154.07bc | 1213 | 996 | 271 | 10322 | 1.40 ±0.16b |
| | RN180 | 289.53 ±14a | 0.18 ±0.07ab | 171.54ab | 1213 | 1494 | 279 | 9679 | 1.25 ±0.09bc |
| | RN240 | 295.46 ±38a | 0.24 ±0.14ab | 185.50ab | 1213 | 1992 | 297 | 10321 | 1.20 ±0.08cd |
| | RN300 | 324.47 ±72a | 0.35 ±0.25a | 196.87a | 1213 | 2490 | 293 | 11550 | 1.38 ±0.21bc |

[a]Definitions of treatment codes are given in the footnotes of Table 1.

[b]Mean ±SD; different letters within same column indicate a significant difference at $p < 0.05$.

**Table 5.** The seasonal average reactive N (Nr) losses and reactive N intensity (NrI) for the two rice-growing seasons from 2013 to 2014 in the Taihu Lake region

| Treatment[a] | $NH_3$ volatilization | N runoff | N leaching | $N_2O$ emission | $NO_X$ emission | Total Nr losses | NrI |
|---|---|---|---|---|---|---|---|
| | kg N ha$^{-1}$ | | | | | | g N kg$^{-1}$ |
| RN0 | 0.64 | 5.39 | 1.44 | 0.07 | 3.96 | 11.50 | 2.14 |
| RN120 | 21.04 | 10.30 | 2.24 | 0.12 | 5.62 | 39.32 | 5.36 |
| RN180 | 31.24 | 14.25 | 2.80 | 0.21 | 6.44 | 54.93 | 7.12 |
| RN240 | 41.44 | 19.70 | 3.50 | 0.27 | 7.26 | 72.17 | 8.42 |
| RN300 | 51.64 | 27.24 | 4.37 | 0.38 | 8.07 | 91.69 | 10.92 |

[a]Definitions of treatment codes are given in the footnotes of Table 1.

**Table 6.** The  economic indicators (two-season average) for rice production of the  growing seasons from 2013 to 2014 in the Taihu Lake region (unit: ¥ha$^{-1}$)

| Treatment[a] | Yield income[b] | Input costs[c] | Farmer's income[d] | Environmental costs[e] | |
|---|---|---|---|---|---|
| | | | | GHG emissions | Nr releases |
| RN0 | 16125 | 4493 | 11632 | 1143 | 71 |
| RN120 | 22020 | 6104 | 15916 | 1363 | 376 |
| RN180 | 23130 | 6542 | 16588 | 1278 | 535 |
| RN240 | 25725 | 7277 | 18448 | 1362 | 700 |
| RN300 | 25185 | 7385 | 17800 | 1525 | 874 |

[a]Definitions of treatment codes are given in the footnotes of Table 1.

[b]Yield income = rice yield × rice price.

[c]Input costs denote the economic input of purchasing various agricultural materials and hiring labours.

[d]Farmer's income = Yield income − Input costs.

[e]Environmental costs denoted the sum of the acidification costs, eutrophication costs and global warming costs incurred by GHG emissions and Nr releases. The cost prices of GHG and Nr releases are as followed: GHG emission, 132 ¥t$^{-1}$ $CO_2$ eq (Xia et al., 2014); $NH_3$ volatilization, 13.12 ¥kg$^{-1}$ N; N leaching, 6.12 ¥kg$^{-1}$ N; N runoff, 3.64 ¥kg$^{-1}$ N; $NO_X$ emission, 8.7 ¥kg$^{-1}$ N (Xia and Yan, 2011).

**Figure captions**

**Fig. 1. Seasonal variations in the daily precipitation and the temperature during the two rice-growing seasons of (a) 2013 and (b) 2014.**

**Fig.2. Relationship between N fertilizer application rate and average rice yield over the two rice-growing seasons of 2013 and 2014 in the Taihu Lake region.** The vertical bars represent standard errors.

**Fig.3. Seasonal variations in (a) CH$_4$ and (b) N$_2$O fluxes during the two rice-growing seasons from 2013 to 2014 in the Taihu Lake region.** The arrow indicates N fertilizer application. The vertical bars represent standard errors.

**Fig.4. Relationship between N fertilizer application rate and (a) NH$_3$ volatilization, (b) N runoff, (c) N leaching and (d) N$_2$O emissions for rice production in the Taihu Lake region.** These relationships were obtained through a meta-analysis.

**Fig.5. Seasonal average total environmental costs incurred by greenhouse gas (GHG) emissions and reactive N (Nr) losses for rice production in Taihu Lake region.**

[Figure]

**Fig.1**

[Figure]

**Fig.2**

[Figure]

**Fig.3**

[Figure]

**Fig.4**

[Figure]

**Fig.5**

---

## Author Comment (AC5) · 1 Jun 2016

[revised manuscript text omitted]

Besides such substantial releases of Nr and GHG in a direct way, indirect releases during the production of various agricultural materials used for farming operations, are also not ignorable, due to higher input rates of these materials caused by intensive cultivation (Zhang et al., 2013;

Cheng et al., 2014). This warrants the need for life-cycle assessment (LCA) of GHG emissions and Nr releases with respect to rice production in this region.

Considerable environmental costs can be caused by the direct and indirect releases of GHG

and Nr from rice production in the TLR, for instance, in the form of global warming, water eutrophication, or soil acidification (Ju et al., 2009; Xia and Yan, 2011; Xia and Yan, 2012).

Previous studies have proven that environmental costs assessment could provide guidance for emerging policy priorities in mitigating certain GHG or Nr species, after quantifying both their release amounts and damage costs to ecosystems (Gu et al., 2012). However, few studies have attempted to evaluate the total GHG and Nr releases and the associated environmental costs from rice production, with high inputs of N fertilizer and crop straw.

In the present study, we conducted two years of simultaneous measurements of $CH_4$ and nitrous oxide ($N_2O$) emissions from a rice-wheat cropping system in the TLR to evaluate the impacts of simultaneous inputs of crop straw and N fertilizer on (1) net global warming potential (NGWP) and GHG intensity (GHGI), (2) total Nr losses and Nr intensity (NrI), (3) environmental costs incurred by these GHG and Nr releases associated with rice production, from the perspective of LCA.

**2 Materials and methods**

**2.1 Experimental site**

The field experiment was conducted in a paddy rice field at Changshu Agroecological

Experimental Station (31°32′93″N, 120°41′88″E) in Jiangsu province, which is located in the TLR

of China where the cropping system is primarily dominated by summer rice (*Oryza sativa* L.) and winter wheat (*Triticum aestivum* L.) rotation. The climate of the study area is subtropical monsoon, with a mean air temperature of 16.1 ℃ and mean annual precipitation of 990 mm, of which 60–70% occurs during the rice-growing season. The daily mean temperature and precipitation during two rice-growing seasons from 2013 to 2014 are shown in Fig.1. The paddy soil is classified as Anthrosol, which develops from lacustrine sediments. The topsoil (0–20cm)

has a pH of 7.68 ($H_2O$). The bulk density is 1.16 g $cm^{-3}$, the organic C content is 20.1 g C $kg^{-1}$, the total N is 1.98 g $kg^{-1}$, the available P is 11.83 mg $kg^{-1}$ and the available K is 126 mg $kg^{-1}$.

**2.2 Experimental design and field management**

The field experiment included five treatments of different N fertilization rates for rice production: 0 (RN0), 120 (RN120), 180 (RN180), 240 (RN240) and 300 kg N $ha^{-1}$ (RN300, traditional N applied rate in the TLR). Consistent with local practices, wheat straws were harvested, chopped and fully incorporated into soil before rice transplantation in all treatments (Table 1). All of the treatments are laid out in a randomized block design with three replicates, and each plot covered an area of 3 m × 11 m (33 $m^2$).

Rice is transplanted in the middle of June and harvested at the beginning of November. N

fertilizer (in the form of urea) was split into three parts during the rice-growing season: 40% as basal fertilizer, 30% as tiller fertilizer, and 30% as panicle fertilizer. Phosphorus (in the form of calcium superphosphate) and potassium (in the form of potassium chloride) were applied as basal fertilizer at rates of 30 kg $P_2O_5$ $ha^{-1}$ and 60 kg $K_2O$ $ha^{-1}$, respectively. All basal fertilizers were thoroughly incorporated into the soil through plowing, while topdressing fertilizers were applied evenly to the soil surface. According to local practices, the water regime of 'flooding-midseason drainage-flooding-moist but non-waterlogged by intermittent irrigation' was adopted. Details of the specific agricultural management practices for rice production are provided in Table 1.

**2.3 Gas fluxes and topsoil organic carbon sequestration rate**

The $CH_4$ and $N_2O$ fluxes during the rice-growing seasons of 2013 and 2014 were measured using a static chamber and gas chromatography technique. Details of the procedures used for sampling and analysis the gases were described in Xia et al. (2014).

Generally, it takes long-term observations over years to decades before the soil organic carbon (SOC) change is detectable (Yan et al., 2011). The SOC content changes of short-term field experiment couldn't be correctly measured, due to the high variability of SOC during the preliminary several years of the experiment. Therefore, we used the following relationship between the straw input rate (kg C $ha^{-1}$ $yr^{-1}$) and SOC sequestration rate (SOCSR, kg C $ha^{-1}$ $yr^{-1}$), obtained through an on-going long-term straw application experiment in the same region, to calculate the SOCSR in this study (Xia et al., 2014):

$\quad$ SOCSR = Straw input rate $\times 0.0603 + 31.39$ ($R^2 = 0.92$); $\qquad$ (1)

This on-gonging long-term field experiment is also taking place at the Changshu

Agroecological Experimental Station (since 1990), which includes three straw application levels:

0, 4.5 t, and 9.0 t dry-weight $ha^{-1}$ $yr^{-1}$. The equation (1) was established based on the results of

22-year observation (Xia et al., 2014). Same agricultural management practices were applied to the on-going long-term experiment and the experiment of this study.

**2.4 Net global warming potential and greenhouse gas intensity**

The net global warming potential (NGWP, kg $CO_2$ eq $ha^{-1}$) and greenhouse gas intensity (GHGI, kg $CO_2$ eq $kg^{-1}$) of rice production in the TLR was calculated using the following equations:

$NGWP = \sum_{i=1}^{m} AI_{ico_2} + CH_4 \times 25 + N_2O \times 44/28 \times 298 - SOCSR \times 44/12;$     (2)

$GHGI = NGWP/rice\ yield;$         (3)

Here, $AI_{ico_2}$ denotes the GHG emissions from the production and transportation of agricultural inputs, which are calculated by multiplying their application rates by their individual GHG

emission factors, such as synthetic fertilizers, diesel oil, electricity and pesticides (Liang, 2009;

Zhang et al., 2013). $CH_4$ (kg $CH_4$ $ha^{-1}$), $N_2O$ (kg N $ha^{-1}$) and SOCSR (kg C $ha^{-1}$ $yr^{-1}$) represent the $CH_4$ and $N_2O$ emissions from rice production, and the SOC sequestration rate, respectively.

**2.5 Total Nr losses and Nr intensity**

The total Nr losses (kg N $ha^{-1}$) and Nr intensity (NrI, g N $kg^{-1}$) were calculated using the following equations:

$Total\ Nr\ losses = \sum_{i=1}^{m} AI_{iN_r} + (NH_3 + N_2O + N_{Leaching} + N_{Runoff})_{rice};$   (4)

$NH_3\ volatilization = 0.17 \times N_{rate} + 0.64;$     (5)

$N\ runoff = 5.39 \times Exp\ (0.0054 \times N_{rate});$     (6)

$N\ leaching = 1.44 \times Exp\ (0.0037 \times N_{rate});$     (7)

$NrI = (1000 \times Total\ Nr\ losses)\ /rice\ yield\ ;$     (8)

Here, $AI_{iNr}$ denotes the Nr lost (mainly through $N_2O$ and $NO_X$ emissions) from the production and transportation of agricultural inputs (Liang, 2009; Zhang et al., 2013), while

'$(NH_3+N_2O+N_{Leaching}+N_{Runoff})_{rice}$' represents the $NH_3$ volatilization, $N_2O$ emissions, N leaching and runoff during the rice-growing season. $N_{rate}$ represents the N fertilizer application rate. Nr empirical models (Equation 5, 6, 7) derived from a meta-analysis of published literature concerning Nr losses from rice production in the TLR. Specific details regarding this literature survey are provided in Appendix A.

**2.6 Total environmental costs incurred by GHG and Nr releases and farmer's income**

The total environmental costs ($¥ha^{-1}$) incurred by GHG and Nr releases and farmer's income from rice production in the TLR were calculated based on the following equations:

Environmental costs $= \sum_{i=1}^{n}(Nr_iA \times DC_i) + CO_2A \times DC_{CO2};$      (9)

Farmer's income $=$ rice yield $\times$ rice price $-$ input costs;      (10)

$Nr_iA$ (kg N) represents the release amounts of certain Nr species (i), and $DC_i$ ($¥kg^{-1}$ N) denotes the damage cost (DC) per kg of certain Nr (i). $CO_2A$ (ton) and $DC_{CO2}$ ($¥ton^{-1}$) represent the $CO_2$

emissions amount and global warming cost of $CO_2$, respectively. $N_2O$ is both a GHG and Nr species, but its environmental cost was calculated as a GHG here. Because the cost of $N_2O$

emission as Nr species is to damage human health (Gu et al., 2012), but the effects of Nr losses on the damage costs of human health were not included in this study. The environmental costs mainly refer to the global warming incurred by GHG emissions, soil acidification incurred by $NH_3$ and

$NO_X$ emissions, and aquatic eutrophication caused by $NH_3$ emissions, N leaching and runoff (Xia and Yan, 2012).

**2.7 Nitrogen use efficiency and $N_2O$ emission factor**

Nitrogen use efficiency (NUE) and $N_2O$ emission factor ($EF_d\%$) were respectively calculated by the following equations (Ma et al., 2013; Yan et al., 2014):

NUE $= (U_N - U_0)/F_N;$      (11)

$EF_d\% = (E_N - E_0)/F_N;$      (12)

Here, $U_N$ is the plant N uptake (kg ha$^{-1}$) measured in aboveground biomass at physiological maturity in the N fertilization treatments, while $U_0$ is the N uptake measured in aboveground biomass in the treatment without N fertilizer addition (RN0). $E_N$ denotes the cumulative $N_2O$

emissions in the N fertilization treatments, while $E_0$ denotes the $N_2O$ emissions in the RN0. $F_N$

represents the application rate of N fertilizer. The N uptake in straw and grain was analysed via concentrated sulfuric acid digestion and the Kjeldahl method (Zhao et al., 2015).

**2.8 Statistical analysis**

Differences in seasonal $CH_4$, $N_2O$ emissions and rice yield of the two rice-growing seasons from 2013 to 2014 affected by fertilizer treatments, year and their interaction were examined by using a two-way analysis of variance (ANOVA) (Table 2). The grain yield, seasonal $CH_4$ and $N_2O$

emissions, SOCSR and GHGI of different treatments were tested by analysis of variance, and mean values were compared by least significant difference (LSD) at the 5% level. All these analyses were carried out using the SPSS (Version 19.0, USA).

**3 Results and discussion**

**3.1 Rice yield and NUE**

The two-way ANOVA analyses indicated that the rice grain yields were significantly affected by the year and fertilizer treatment (Table 2). The farmer's practice plot (RN300) had an average rice grain yield of 8395 kg ha$^{-1}$, with an NUE of 31.35%, over the two growing seasons from

2013 to 2014. Compared with RN300, reducing the N fertilizer rate by 20% (RN240) slightly improved the grain yield and NUE to 8576 kg ha$^{-1}$ and 34.58%, respectively. Further N reduction, without additional agricultural managements, could decrease the rice yield by 8.15% (RN180) and

15.18% (RN120) (Table 3). The response of rice yield to the synthetic N application rate in our study successfully fitted a quadratic model (Fig.2), as has been reported in previous studies (Xia and Yan, 2012; Cui et al., 2013a). Reducing N application reasonably, therefore, is considered essential to reduce environmental costs, without sacrificing grain yield (Chen et al., 2014). Our study showed that lowering the N input adopted by local farmer (300 kg N ha$^{-1}$) by 20% could still enhance the grain yield and NUE. However, a further reduction of N 40% (RN180) would largely undermine the rice yield (Table 3).

Further reduction in N fertilizer may be achieved with improvements of agricultural managements, Ju et al. (2009) reported that, based on knowledge-based N managements, such as optimizing N fertilizer source, rate, timing and place (in accordance with crop demand), rice grain yield in the TLR was not significantly affected by a 30–60% N saving, while various Nr losses would endure a two-fold curbing. Similarly, Zhao et al. (2015) found that the NUE could be improved from 31% to 44%, even under a N reduction of 25% for rice production in the TLR, through the implementation of integrated soil-crop system managements. In the present study, the

NUE was improved by 10% via a 20% N reduction, but it still falls behind the NUE values in the studies which received knowledge-based N managements. Previous studies have proven that straw incorporation exerted little impacts on grain yield. For instance, a meta-analysis conducted by

Singh et al. (2005) have found that incorporation of crop straw produced no significant trend in improving crop yield in rice-based cropping systems. Moreover, based on a long-term straw incorporation experiment established since 1990 in the TLR, Xia et al. (2014) have reported that long-term incorporation of wheat straw only increased the rice yield by 1%. Therefore, in the present study, the effects of straw incorporation on rice yield were considered as inappreciable.

**3.2 CH$_4$, N$_2$O emissions and SOSCR**

Over the two rice-growing seasons from 2013 to 2014, all treatments showed similar patterns of $CH_4$ fluxes, albeit with large inter-annual variation (Fig.3a). The seasonal average $CH_4$

emissions from all plots showed no significant difference, ranging from 289.53 kg $CH_4$ $ha^{-1}$ in the

RN180 plot to 334.61 kg $CH_4$ $ha^{-1}$ in the RN120 plot (Table 4), much higher than observations conducted in the same region (Zou et al., 2005; Ma et al., 2013). This phenomenon can be attributed to the larger amounts of straw incorporation in this study (Table 1). Relative to the

RN300 plot, $CH_4$ emissions from the RN240 plot decreased by 8% and 10%, during the rice-growing season of 2013 and 2014, respectively, although this effect was not statistically significant (Table 4).

Many studies have shown a clear linear relationship between $CH_4$ emissions and the amounts of applied organic matter (OM) (Shang et al., 2011; Xia et al., 2014). It is possible that the linear response of $CH_4$ emissions to OM inputs can become flat or even unobvious (Fig.S1), when the

OM applied rates among different treatments were insignificant different (Table S1). It is unsurprising that no obvious relationship between $CH_4$ emissions and N fertilizer application rates was observed in this study (Fig.S1), because the effects of N fertilization on $CH_4$ production, transportation and oxidation are complex. For instance, N fertilization can provide methanogens with more carbon substrates in the rhizosphere of plants by stimulating the growth of rice biomass, thus promoting $CH_4$ production and transportation (Zou et al., 2005; Banger et al., 2012). On the other side, N enrichment could also enhance the activities of methanotrophs, therefore enhancing

$CH_4$ oxidation (Xie et al., 2010; Yao et al., 2012).

The $N_2O$ fluxes were sporadic and pulse-like, and these fluxes showed large variations between different seasons, and the majority of the $N_2O$ peaks occurred after the application of N

fertilizer (Fig.3b). The two-way ANOVA analyses indicated that the seasonal $N_2O$ emissions were significantly affected by the year, the fertilizer treatment, and their interactions during the rice-growing seasons (Table 2). The average $N_2O$ emission, during the two rice-growing seasons, ranged from 0.05 kg N ha$^{-1}$ for the RN0 to 0.35 kg N ha$^{-1}$ for the RN300 (Table 4), which increased exponentially as the N fertilizer rate increased; this highlights that the reduction of N

fertilizer rate is an effective approach to reduce the $N_2O$ emissions (Zou et al., 2005; Zhang et al.,

2012). The average $N_2O$ emission factors varied between 0.03% and 0.1%, with an average of

0.07%, which is comparable with previous studies (0.05–0.1%) conducted in the same region (Ma et al., 2013; Zhao et al., 2015).

The rice paddies have witnessed an increase in the SOC stock as a result of straw incorporation (Table 4). The estimated topsoil (0–20cm) SOCSR varied from 0.13 t C ha$^{-1}$ yr$^{-1}$ for the RN0 plot to 0.197 t C ha$^{-1}$ yr$^{-1}$ for the RN300 plot. The empirical model established through a long-term straw incorporation study in the same region was employed to evaluate the SOCSR in this study, which likely brought uncertainty into the results of this study. Under the same agricultural managements, soil and climatic conditions, cropping systems and straw types, it is reasonable to believe that the rates of straw C stabilizing into SOC (i.e. conversion efficiency of crop residue C into SOC) are similar between these two experiments (Mandal et al., 2008). It is reported that the conversion rates of crop straw to SOC in two main wheat/maize production regions in China, which have similar climatic conditions and agricultural practices, were very close, at 40.524 versus 40.607 kg SOC-C t$^{-1}$ dry-weight straw (Lu et al., 2009). Moreover, the current estimated SOCSR for rice production in the TLR (0.197 t C ha$^{-1}$), is comparable to the estimation of 0.17 t C ha$^{-1}$ yr$^{-1}$ from Ma et al. (2013) in a study based on a paddy field experiment with OM incorporation in the same region. Therefore, we hold the opinion that the above SOCSR

calculation method is appropriate, and the uncertainty incurred by this method unlikely affects the main conclusions of this study.

The magnitude of the SOC increase is variable depending on the straw incorporation method, the degree of tillage, the cropping systems and etc. (Yan et al., 2011; Huang et al., 2013). Liu et al.

(2014) suggested that straw incorporation in rice-based cropping systems requires an overall consideration, due to the direct incorporation promoting substantial $CH_4$ emissions. When converting to $CO_2$ eq, the SOCSR only offsets the $CH_4$ emissions by 6.2–9.2% in this study (Table

4). This proportion is expected to increase provided that appropriate straw incorporation method (e.g., compost straw before incorporation) and conservative-tillage are adopted. Moreover, previous studies have shown that the combined adoption of conservative-tillage system with straw return had large advantages in increasing SOC stocks while reducing $CH_4$ emissions (Zhao et al.,

2015a; Zhao et al., 2015b).

**3.3 NGWP and GHGI**

The average NGWP for all treatments varied from 8656 to 11550 kg $CO_2$ eq $ha^{-1}$ (Table 4).

$CH_4$ emissions dominated the NGWP in all treatments, with the proportion ranging from 70.23%

to 88.56%, while synthetic N fertilizer production was the secondary contributor (Table 4). In addition, SOC sequestration offset the positive GWP by 5.18–6.18% in the fertilization treatments.

Compared to conventional practice (RN300), the NGWP in the 20% reduction N practice (RN240)

decreased by 10.64%. Therein, 6.28% came from $CH_4$ reduction and 4.31% from N production savings (Table 4). The GHGI of rice production ranged from 1.20 (RN240) to 1.61 (RN0) kg $CO_2$

eq $kg^{-1}$, which is higher than previous estimation of 0.24–0.74 kg $CO_2$ eq $kg^{-1}$ for rice production in other rice-upland crop rotation systems (Qin et al., 2010; Ma et al., 2013). Moreover, the GHGI

of current rice production in the TLR (RW300) was estimated to be 1.45 times that of the national average value estimated by Wang et al. (2014a), at 1.38 versus 0.95 kg $CO_2$ eq $kg^{-1}$.

Such phenomenon was attributed to the following reasons. First, compared to above studies, current higher amounts of direct straw incorporation (2.9–6.2 Mg dry matter $ha^{-1}$), before rice transplantation in the TLR, triggered substantial $CH_4$ emissions (290–335 kg $CH_4$ $ha^{-1}$). Crop residue incorporation is regarded as a win-win strategy to benefit food security and mitigate climate change, due to the fact that it possesses a large potential for carbon sequestration (Lu et al.,

2009). However, the GWP of straw-induced $CH_4$ emissions was reported to be 3.2–3.9 times that of the straw-induced SOCSR, which indicates direct straw incorporation in paddy soils worsens rather than mitigates climate changes, in terms of GWP (Xia et al., 2014). The SOC sequestration induced by straw incorporation only offset the positive GWP by 5.2–6.2% in this study. Sensible methods of straw incorporation should therefore be developed to reduce the substantial $CH_4$

emissions without compromising the build-up of SOC stock in the TLR.

Second, the high N application rate (300kg N $ha^{-1}$) in the TLR combined with the large emission factor of N fertilizer production, 8.3 kg $CO_2$ eq $kg^{-1}$ N (Zhang et al., 2013), marked the sector of N fertilizer production as the secondary contributor to the GHGI (Table 4); this sector, however, was not involved in above-mentioned studies. Compared to local farmer's practices (RN300), reducing the N rate by 20% (RN240) lowered the GHGI by 13%, under the condition of straw incorporation, although this effect was not statistically significant (Table 4). Compared to

RN240, however, further reduction of N rate (RN180 or RN120) increased the GHGI, due to the fact that rice yield was considerably reduced under excessive N reduction. Therefore, the joint application of reasonable N reduction and judicious method of straw incorporation would be promising in reducing the GHGI for rice production in the TLR, in consideration of the current situation of simultaneous high inputs of N fertilizer and wheat straw.

**3.4 Various Nr losses and NrI**

The results of the meta-analysis indicated that $N_2O$ emissions, as well as N leaching and runoff, increased exponentially with an increase in N application rate (Fig.4b-d, $P < 0.01$), while the response of $NH_3$ volatilization to N rates fitted the linear model best (Fig.4a, $P < 0.01$). The estimated total Nr losses for all treatments varied from 39.3 to 91.7 kg N $ha^{-1}$ in the fertilization treatments (Table 5), accounting for 30.1–32.8% of N application rates. $NH_3$ volatilization dominated the NrI, with the proportion ranging from 53.5% to 57.4%, mainly because of the current fertilizer application method (soil surface broadcast) and high temperatures in the field (Zhao et al., 2012b; Li et al., 2014). N runoff was the second most important contributor (Table 5).

Using $^{15}N$ micro-plots combined with three-year field measurements, Zhao et al. (2012b) reported that the total Nr losses from rice production in the TLR, under an N rate of 300 kg N $ha^{-1}$, were 98

kg N $ha^{-1}$, which is comparable with our estimation of 91.69 kg N $ha^{-1}$ in the RN300 plot.

Similarly, Xia and Yan (2011) estimated the Nr losses for life-cycle rice production in this region to be around 90 kg N $ha^{-1}$. The high proportion (30.1–32.8%) of the applied N fertilizer released as Nr from rice production in the TLR, highlights the need to adopt reasonable N managements to increase the plant N uptake and reduce Nr losses (Ju et al., 2009).

The NrI of rice production in different plots varied between 2.14 g N $kg^{-1}$ (RN0) and 10.92 g

N $kg^{-1}$ (RN300), which increased significantly as the N fertilizer rate increased (Table 5). The NrI

for rice production in the TLR was estimated to be 10.92 g N $kg^{-1}$ (RN300), which is 68% higher than the national average value estimated by Chen et al. (2014), as a result of higher N fertilizer input in the TLR. Under the condition of straw incorporation, reducing N application rate by 20%

pulled the NrI down to 8.42 g N kg$^{-1}$ (RN240) (Table 5). Additional N reduction could further lower the NrI, but the rice yield would be compromised largely (Table 3). Previous studies have proven that direct incorporation of crop straw had insignificant effects on various Nr releases (Xia et al., 2014). Because the majority of N contented in the crop straw is not easily degraded by microorganisms in a short-term period, and can be stabilized in soil in a long-term period, rather than being released as various Nr (Huang et al., 2004; Xia et al., 2014). For instance, a meta-analysis, integrating 112 scientific assessments of the crop residue incorporation on the $N_2O$

[revised manuscript text omitted]

'Carbon Farming Initiative' in Australia (Lam et al., 2013), have been launched in China, an ecological compensation incentive mechanism should be established by governments. This should be a national subsidy program with a special compensation and award fund to cover the extra mitigation costs induced by the adoption of knowledge-based mitigation managements for farmers (Xia et al., 2016). Such a program would provide farmers with a tangible incentive, thus guiding them towards gradually adopting the mitigation managements, which could effectively curb GHG

emissions and Nr losses, but likely exert little positive effects on improving their net economic return (Xia et al., 2014). Examples include the composing of crop straws aerobically, or their use to produce biochar before incorporation (Xie et al., 2013), and encouraging the application of deep placement of N fertilizer (Wang et al., 2014b), as well as the application of enhanced-efficiency N

fertilizers during the rice-growing season (Akiyama et al., 2010).

**4 Conclusions**

Our results demonstrated that producing rice yield in the TLR released substantial GHG and

Nr, which largely attributed to the current direct straw incorporation and excessive N fertilizer inputs. $CH_4$ emissions and $NH_3$ volatilization dominated the GHG and Nr releases, respectively.

Reducing N application rate by 20% from the tradition level (300 kg N $ha^{-1}$) could effectively decrease the GHG emissions, Nr releases and the damage costs to the environment, while increased the rice yield and improved farmer's income simultaneously. Agricultural managements, such as making straw decompose aerobically before its incorporation and optimizing the application method of N fertilizer, showed large potentials to further reduce the GHG (e.g., $CH_4$

emission) and Nr releases (e.g., $NH_3$ volatilization) from rice production in this region. Further studies are needed to evaluate the comprehensive effects of these managements on GHG

emissions, Nr releases and farmer's economic returns.

**Acknowledgements**

[revised manuscript text omitted]

  system management, Agric. Ecosyst. Environ., 164, 209-219, 2013.

Mandal, B., Majumder, B., Adhya, T., Bandyopadhyay, P., Gangopadhyay, A., Sarkar, D., Kundu,

  M., Choudhury, S.G., Hazra, G., Kundu, S.: Potential of double-cropped rice ecology to

  conserve organic carbon under subtropical climate, Global Change Biol., 14, 2139-2151,

2008.

Nayak, D., Saetnan, E., Cheng, K., Wang, W., Koslowski, F., Cheng, Y., Zhu, W.Y., Wang, J., Liu,

J., Moran, D.: Management opportunities to mitigate greenhouse gas emissions from

Chinese agriculture, Agric. Ecosyst. Environ., 209, 108-124, 2015.

Qin, Y., Liu, S., Guo, Y., Liu, Q., Zou, J.: Methane and nitrous oxide emissions from organic and conventional rice cropping systems in Southeast China, Biol. Fertil. Soils, 46,

825-834, 2010.

Shan, J., Yan, X.Y.: Effects of crop residue returning on nitrous oxide emissions in agricultural soils, Atmos. Environ., 71, 170-175, 2013.

Shang, Q., Yang, X., Gao, C., Wu, P., Liu, J., Xu, Y., Shen, Q., Zou, J., Guo, S.: Net annual global warming potential and greenhouse gas intensity in Chinese double rice-cropping systems:

a 3-year field measurement in long-term fertilizer experiments, Global Change Biol., 17,

2196-2210, 2011.

Singh, Y., Singh, B., Timsina, J.: Crop residue management for nutrient cycling and improving soil productivity in rice-based cropping systems in the tropics, Adv. Agron., 85, 269-407,

2005.

Wang, W., Guo, L., Li, Y., Su, M., Lin, Y., De Perthuis, C., Ju, X., Lin, E., Moran, D.: Greenhouse gas intensity of three main crops and implications for low-carbon agriculture in China,

Climatic Change, 128, 57-70, 2014a.

Wang, W., Koslowski, F., Nayak, D.R., Smith, P., Saetnan, E., Ju, X., Guo, L., Han, G., de

Perthuis, C., Lin, E., Moran, D.: Greenhouse gas mitigation in Chinese agriculture:

Distinguishing technical and economic potentials, Global Environ. Chang., 26, 53-62,

2014b.

Woolf, D., Amonette, J.E., Street-Perrott, F.A., Lehmann, J., Joseph, S.: Sustainable biochar to

  mitigate global climate change, Nat. Commun., 1, 56, 2010.

Xia, L., Wang, S., Yan, X.: Effects of long-term straw incorporation on the net global warming

  potential and the net economic benefit in a rice-wheat cropping system in China, Agric.

  Ecosyst. Environ., 197, 118-127, 2014.

Xia, L., Ti, C., Li, B., Xia, Y., Yan, X.:Greenhouse gas emissions and reactive nitrogen

  releases during the life-cycles of staple food production in China and their mitigation

  potential, Sci. Total Environ., 556, 116-125, 2016.

Xia, Y., Yan, X.: Life-cycle evaluation of nitrogen-use in rice-farming systems: implications for

  economically-optimal nitrogen rates, Biogeosciences, 8, 3159-3168, 2011.

Xia, Y., Yan, X.: Ecologically optimal nitrogen application rates for rice cropping in the Taihu

  Lake region of China, Sustain. Sci., 7, 33-44, 2012.

Xie, B., Zheng, X., Zhou, Z., Gu, J., Zhu, B., Chen, X., Shi, Y., Wang, Y., Zhao, Z., Liu, C.:

  Effects of nitrogen fertilizer on $CH_4$ emission from rice fields: multi-site field observations,

  Plant Soil, 326, 393-401, 2010.

Xie, Z., Xu, Y., Liu, G., Liu, Q., Zhu, J., Tu, C., Amonette, J.E., Cadisch, G., Yong, J.W., Hu, S.:

  Impact of biochar application on nitrogen nutrition of rice, greenhouse-gas emissions and

  soil organic carbon dynamics in two paddy soils of China, Plant Soil, 370, 527-540, 2013.

Yan, X., Akiyama, H., Yagi, K., Akimoto, H.: Global estimations of the inventory and mitigation

  potential of methane emissions from rice cultivation conducted using the 2006

  Intergovernmental Panel on Climate Change Guidelines, Global Biogeochem. Cycles, 23,

2009.

Yan, X., Cai, Z., Wang, S., Smith, P.: Direct measurement of soil organic carbon content change in the croplands of China, Global Change Biol., 17, 1487-1496, 2013.

Yan, X., Ti, C.,Vitousek, P., Chen, D., Leip, A., Cai, Z., Zhu, Z.: Fertilizer nitrogen recovery efficiencies in crop production systems of China with and without consideration of the residual effect of nitrogen, Environ. Res. Lett., 9, 095002, 2014.

Yao, Z., Zheng, X., Dong, H., Wang, R., Mei, B., Zhu, J.: A 3-year record of $N_2O$ and $CH_4$

emissions from a sandy loam paddy during rice seasons as affected by different nitrogen application rates, Agric. Ecosyst. Environ., 152, 1-9, 2013.

Zhang, F., Cui, Z., Chen, X., Ju, X., Shen, J., Chen, Q., Liu, X., Zhang, W., Mi, G., Fan, M.:

Integrated nutrient management for food security and environmental quality in China, Adv.

Agron., 116, 1-40, 2012.

Zhang, W., Dou, Z., He, P., Ju, X., Powlson, D., Chadwick, D., Norse, D., Lu, Y., Zhang, Y., Wu,

L.: New technologies reduce greenhouse gas emissions from nitrogenous fertilizer in

China, Proc. Natl. Acad. Sci. U.S.A., 110, 8375-8380, 2013.

Zhao, M., Tian, Y., Ma, Y., Zhang, M., Yao, Y., Xiong, Z., Yin, B., Zhu, Z.: Mitigating gaseous nitrogen emissions intensity from a Chinese rice cropping system through an improved management practice aimed to close the yield gap, Agric. Ecosyst. Environ., 203, 36-45,

2015.

Zhao, X., Zhou, Y., Min, J., Wang, S., Shi, W., Xing, G.: Nitrogen runoff dominates water nitrogen pollution from rice-wheat rotation in the Taihu Lake region of China, Agric. Ecosyst.

Environ., 156, 1-11, 2012a.

Zhao, X., Zhou, Y., Wang, S., Xing, G., Shi, W., Xu, R., Zhu, Z.: Nitrogen balance in a highly fertilized rice-wheat double-cropping system in southern China, Soil Sci. Soc. Am. J., 76,

1068-1078, 2012b.

Zhao, X., Liu, S.L., Pu, C., Zhang, X.Q., Xue, J.F., Zhang, R., Wang, Y.Q., Lal, R., Zhang, H.L.,

Chen, F.:Methane and nitrous oxide emissions under no-till farming in China: a meta-analysis. Global Change Biol., 22, 1372-1384, 2015a.

Zhao, X., Zhang, R., Xue, J.F., Pu, C., Zhang, X.Q., Liu, S.L., Chen, F., Lal, R., Zhang, H.L.:

Management-induced changes to soil organic carbon in China: A meta-analysis. Adv.

Agron., 134, 1-49, 2015b.

Zou, J., Huang, Y., Jiang, J., Zheng, X., Sass, R.L.: A 3-year field measurement of methane and nitrous oxide emissions from rice paddies in China: Effects of water regime, crop residue, and fertilizer application, Global Biogeochem. Cycles, 19, 2005.

**Table 1.** Field experimental treatments and agricultural management practices during the rice-growing seasons of 2013 and 2014 in the Taihu Lake region

| Treatment[a] | RN0 | RN120 | RN180 | RN240 | RN300 |
|---|---|---|---|---|---|
| Chemical fertilizer application rate ($N:P_2O_5:K_2O$, kg ha$^{-1}$) | 0:30:60 | 120:30:60 | 180:30:60 | 240:30:60 | 300:30:60 |
| Split N application ratio | --- | 4:3:3 | 4:3:3 | 4:3:3 | 4:3:3 |
| Straw application rate (Mg dry matter ha$^{-1}$) | 3.94/2.88[b] | 4.49/4.65 | 4.93/5.18 | 5.33/5.87 | 5.81/6.17 |
| Water regime[c] | F-D-F-M | F-D-F-M | F-D-F-M | F-D-F-M | F-D-F-M |
| Density ($10^4$ plants ha$^{-1}$) | 2.5 | 2.5 | 2.5 | 2.5 | 2.5 |

[a]RN0, RN120, RN180, RN240 and RN300 represent N application rates of 0, 120, 180, 240, 300

kg N ha$^{-1}$, respectively.

[b]3.94/2.88 denote that straw application rates during the rice-growing seasons of 2013 and 2014

are 3.94 and 2.88 Mg dry matter ha$^{-1}$, respectively.

[c]F, flooding; D, midseason drainage; M, moist but non-waterlogged by intermittent irrigation.

**Table 2.** Two-way ANOVA for the effects of fertilizer (F) application and year (Y) on $CH_4$ and

$N_2O$ emissions, and rice grain yields in rice paddies.

| Factor | df | $CH_4$ (kg ha$^{-1}$) | | | $N_2O$ (kg N ha$^{-1}$) | | | Yield (kg ha$^{-1}$) | | |
|---|---|---|---|---|---|---|---|---|---|---|
| | | SS | F | P | SS | F | P | SS | F | P |
| F | 4 | 8739 | 0.79 | 0.55 | 0.33 | 12.46 | < 0.01 | 39297547 | 32.96 | < 0.01 |
| Y | 1 | 4492 | 1.62 | 0.22 | 0.11 | 16.41 | < 0.01 | 2810414 | 9.43 | < 0.01 |
| F×Y | 4 | 2532 | 0.23 | 0.92 | 0.18 | 7.1 | < 0.01 | 750639 | 0.63 | 0.65 |
| Model | 9 | 15763 | 0.63 | 0.77 | 0.62 | 10.52 | < 0.01 | 42858600 | 15.97 | < 0.01 |
| Error | 16 | 20 | | | 0.13 | | | 5962260 | | |

**Table 3.** Rice yield and nitrogen use efficiency (NUE) for the two rice-growing seasons from 2013

to 2014 in the Taihu Lake region

| Year | Treatment[a] | Yield (kg ha$^{-1}$) | NUE (%) |
|---|---|---|---|
| 2013 | RN0 | 4829 ±207 | --- |
| | RN120 | 7079 ±645 | 23.40 |
| | RN180 | 7655 ±601 | 28.12 |
| | RN240 | 8273 ±569 | 33.61 |
| | RN300 | 8029 ±101 | 30.63 |
| 2014 | RN0 | 5919 ±131 | --- |
| | RN120 | 7598 ±1077 | 23.86 |
| | RN180 | 7768 ±570 | 21.19 |
| | RN240 | 8880 ±435 | 35.54 |
| | RN300 | 8761 ±369 | 32.07 |
| Two-year average | RN0 | 5374 ±617d[b] | --- |
| | RN120 | 7339 ±843c | 23.63 |
| | RN180 | 7711 ±527bc | 24.66 |
| | RN240 | 8576 ±562a | 34.58 |
| | RN300 | 8395 ±468ab | 31.35 |

[a]Definitions of the treatment codes are given in the footnotes of Table 1.

[b]Mean±SD; different letters within the same column indicate a significant difference at $p<0.05$.

**Table 4.** The net global warming potential (NGWP) and greenhouse gas intensity (GHGI) for the two rice-growing seasons from 2013 to 2014 in the Taihu Lake region

| Year | Treatment[a] | CH$_4$ emission | N$_2$O emission | SOCSR | Irrigation | N fertilizer production | Others | NGWP | GHGI |
|------|-----------|-----------------|-----------------|-------|------------|-------------------------|--------|------|------|
| | | kg CH$_4$ ha$^{-1}$ | kg N ha$^{-1}$ | kg C ha$^{-1}$ yr$^{-1}$ | | kg CO$_2$ eq ha$^{-1}$ | | | kg CO$_2$ eq kg$^{-1}$ |
| 2013 | RN0 | 306.07 ±41[b] | 0.08 ±0.01 | 129.58 | 1170 | 0 | 217 | 8601 | 1.78 |
| | RN120 | 317.26 ±92 | 0.10 ±0.01 | 154.07 | 1170 | 996 | 265 | 9845 | 1.39 |
| | RN180 | 287.8 ±12 | 0.13 ±0.01 | 171.54 | 1170 | 1494 | 277 | 9568 | 1.25 |
| | RN240 | 273.27 ±36 | 0.14 ±0.06 | 185.50 | 1170 | 1992 | 291 | 9670 | 1.17 |
| | RN300 | 305.13 ±90 | 0.16 ±0.03 | 196.87 | 1170 | 2490 | 285 | 10927 | 1.36 |
| 2014 | RN0 | 307.22 ±47 | 0.02 ±0.05 | 129.58 | 1256 | 0 | 240 | 8711 | 1.47 |
| | RN120 | 351.96 ±28 | 0.09 ±0.02 | 154.07 | 1256 | 996 | 276 | 10805 | 1.42 |

| | | | | | | | | |
|---|---|---|---|---|---|---|---|---|
| | RN180 | 291.25 ±18 | 0.24 ±0.04 | 171.54 | 1256 | 1494 | 280 | 9795 | 1.26 |
| | RN240 | 317.65 ±28 | 0.34 ±0.12 | 185.50 | 1256 | 1992 | 303 | 10972 | 1.24 |
| | RN300 | 343.8 ±61 | 0.53 ±0.21 | 196.87 | 1256 | 2490 | 301 | 12169 | 1.39 |
| Two-year average | RN0 | 306.65 ±39a | 0.05 ±0.05b | 129.58c | 1213 | 0 | 229 | 8656 | 1.61 ±0.25a |
| | RN120 | 334.61±64a | 0.09 ±0.02b | 154.07bc | 1213 | 996 | 271 | 10322 | 1.40 ±0.16b |
| | RN180 | 289.53 ±14a | 0.18 ±0.07ab | 171.54ab | 1213 | 1494 | 279 | 9679 | 1.25 ±0.09bc |
| | RN240 | 295.46 ±38a | 0.24 ±0.14ab | 185.50ab | 1213 | 1992 | 297 | 10321 | 1.20 ±0.08cd |
| | RN300 | 324.47 ±72a | 0.35 ±0.25a | 196.87a | 1213 | 2490 | 293 | 11550 | 1.38 ±0.21bc |

[a]Definitions of treatment codes are given in the footnotes of Table 1.

[b]Mean ±SD; different letters within same column indicate a significant difference at $p<0.05$.

**Table 5.** The seasonal average reactive N (Nr) losses and reactive N intensity (NrI) for the two rice-growing seasons from 2013 to 2014 in the Taihu Lake region

| Treatment[a] | $NH_3$ volatilization | N runoff | N leaching | $N_2O$ emission | $NO_X$ emission | Total Nr losses | NrI |
|---|---|---|---|---|---|---|---|
| | $kg\ N\ ha^{-1}$ | | | | | | $g\ N\ kg^{-1}$ |
| RN0 | 0.64 | 5.39 | 1.44 | 0.07 | 3.96 | 11.50 | 2.14 |
| RN120 | 21.04 | 10.30 | 2.24 | 0.12 | 5.62 | 39.32 | 5.36 |
| RN180 | 31.24 | 14.25 | 2.80 | 0.21 | 6.44 | 54.93 | 7.12 |
| RN240 | 41.44 | 19.70 | 3.50 | 0.27 | 7.26 | 72.17 | 8.42 |
| RN300 | 51.64 | 27.24 | 4.37 | 0.38 | 8.07 | 91.69 | 10.92 |

[a]Definitions of treatment codes are given in the footnotes of Table 1.

**Table 6.** The economic indicators (two-season average) for rice production of the growing seasons from 2013 to 2014 in the Taihu Lake region (unit: ¥ha$^{-1}$)

| Treatment[a] | Yield income[b] | Input costs[c] | Farmer's income[d] | Environmental costs[e] | |
|---|---|---|---|---|---|
| | | | | GHG emissions | Nr releases |
| RN0 | 16125 | 4493 | 11632 | 1143 | 71 |
| RN120 | 22020 | 6104 | 15916 | 1363 | 376 |
| RN180 | 23130 | 6542 | 16588 | 1278 | 535 |
| RN240 | 25725 | 7277 | 18448 | 1362 | 700 |
| RN300 | 25185 | 7385 | 17800 | 1525 | 874 |

[a]Definitions of treatment codes are given in the footnotes of Table 1.

[b]Yield income = rice yield ×rice price.

[c]Input costs denote the economic input of purchasing various agricultural materials and hiring labours.

[d]Farmer's income = Yield income − Input costs.

[e]Environmental costs denoted the sum of the acidification costs, eutrophication costs and global warming costs incurred by GHG emissions and Nr releases. The cost prices of GHG and Nr releases are as followed: GHG emission, 132 ¥t$^{-1}$ CO$_2$ eq (Xia et al., 2014); NH$_3$ volatilization, 13.12 ¥kg$^{-1}$ N; N leaching, 6.12 ¥kg$^{-1}$ N; N runoff, 3.64 ¥kg$^{-1}$ N; NO$_X$ emission, 8.7 ¥kg$^{-1}$ N (Xia and Yan, 2011).

**Figure captions**

**Fig. 1. Seasonal variations in the daily precipitation and the temperature during the two rice-growing seasons of (a) 2013 and (b) 2014.**

**Fig.2. Relationship between N fertilizer application rate and average rice yield over the two rice-growing seasons of 2013 and 2014 in the Taihu Lake region.** The vertical bars represent standard errors.

**Fig.3. Seasonal variations in (a) $CH_4$ and (b) $N_2O$ fluxes during the two rice-growing seasons from 2013 to 2014 in the Taihu Lake region.** The arrow indicates N fertilizer application. The vertical bars represent standard errors.

**Fig.4. Relationship between N fertilizer application rate and (a) $NH_3$ volatilization, (b) N runoff, (c) N leaching and (d) $N_2O$ emissions for rice production in the Taihu Lake region.** These relationships were obtained through a meta-analysis.

**Fig.5. Seasonal average total environmental costs incurred by greenhouse gas (GHG) emissions and reactive N (Nr) losses for rice production in Taihu Lake region.**

[Figure]

**Fig.1**

[Figure]

**Fig.2**

[Figure]

**Fig.3**

[Figure]

**Fig.4**

[Figure]

**Fig.5**

---

## Author Comment (AC6) · 22 Jun 2016

Please see the attached file.

Please also note the supplement to this comment:
http://www.biogeosciences-discuss.net/bg-2015-620/bg-2015-620-AC6-supplement.pdf
* * *

---

## Author Comment (AC9) · 22 Jun 2016

Dear Prof. Richard Conant and Reviewers,

On behalf of my co-authors, thank you very much for your positive and constructive comments on our manuscript. We have carefully studied the comments and have made corrections which we hope to meet with approval. Please see the attached point-by-point responses and the tracked change version of manuscript for your further evaluation. All revised positions mentioned in the responses can be readily found in the attached clear version of manuscript.

**Response to Reviewer's comments:**

**Reviewer 1:**

1. The section on the details of the long-term experiment (lines 145-153) is not necessary.

**Response:** Thanks very much for your suggestion. According to your suggestion, we have deleted the details of the long-term experiment (**please see Line 158-167**).

2. It was a bit confusing because the rates of residue incorporation used in this study (Table 1) were different from those in the long-term experiment (line 146).

**Response:** Thanks for your comment and sorry for our unclear expression. Yes, the rates of residue incorporation used in this study were different from those in the long-term experiment, but we think it is appropriate to use the equation to calculate the SOCSR in this study. We have noticed the uncertainty induced by the SOCSR calculation method and discussed it in the results and discussion part of '**CH$_4$, N$_2$O emissions and SOSCR'.** Moreover, we also presented the reasons why we hold the opinion that the SOCSR calculation method in this study is appropriate, and the uncertainty incurred by this method unlikely affects the main conclusions of this study (**please see Line 306-324**).

3. The relationship between $CH_4$ emission and the amount of organic matter input was not the major focus of the paper. The discussion should be simplified rather than being extended with possible explanations, some of which are speculative.

**Response:** Thanks for your good suggestion. According to your suggestion, we have simplified the relevant discussion (**please see Line 274-285**).

4. At a few other places in the discussion section e.g. lines 278-285 the authors presented their results, and compared the results with others', which was fine but the manuscript would be more informative if the implications of the findings could be explored.

**Response:** Thanks for your good suggestion. According to your suggestion, we have explored the implications of the SOC sequestration of this study (**please see Line 325-335**). We also revised somewhere else, such as **Line 301-303** and **Line 387-389**, to illustrate the implications of our findings.

5. Minor comments:

Line 76: delete "And"

**Response:** Agreed and revised (**please see Line 84**).

Line 112: The scientific name of rice was provided but not for winter wheat

**Response:** Sorry for our carelessness. We have added the scientific name of wheat in the text (**please see Line 121**).

Lines 252-253: What does it mean by "the applied OM rates among different treatments are statistically different"? A statistical test on the independent variables (OM

application rates)?

**Response:** Sorry for our unclear expression. We have deleted this sentence.

Line 311: "was not" instead of "wasn't"

**Response:** Agreed and revised (**please see Line 363**).

Lines 344-346: incomplete sentence

**Response:** Sorry for our unclear expression. We have revised this sentence (**please see Line 398-403**).

Once again, thank you very much for your constructive comments and suggestions.

**Reviewer 2: Specific comments**

1. 1. Abstract: Authors employed the meta-analysis to calculate the various Nr losses. As an important part of this study, the results of the meta-analysis should be simply presented in the abstract. Moreover, it would be better if the abstract is concisely shortened, since some findings in the current version were insignificant, e.g., L34 'while methane emission …..wheat rates increased'.

**Response:** Thanks very much for your comment and suggestion. According to your suggestion, we have presented the main findings of the meta-analysis in the abstract. We have also concisely shortened the abstract (**please see Line 24-54**).

2. L71-72, specify the current water and straw application methods.

**Response:** Thanks for your comment and sorry for our unclear expression. We have specified the water and straw application methods (**please see Line 79-80**).

3. L140 Using the relationship of straw input rate and SOCSR of previous study to calculate the SOC changes is fine, since both of the studies have similar climatic conditions, cropping history and agricultural practices. But the uncertainty should be noticed and can be discussed in the result and discussion part.

**Response:** Thanks for your good suggestion. According to your suggestion, we have noticed the uncertainty induced by the SOCSR calculation method and discussed it in the results and discussion part of '**CH$_4$, N$_2$O emissions and SOSCR'.** Moreover, we also presented the reasons why we hold the opinion that the SOCSR calculation method in this study is appropriate, and the uncertainty incurred by this method unlikely affects the main conclusions of this study (**please see Line 306-324**).

4. L193-205. The environmental cost evaluation is interesting. But, why treated N$_2$O as a GHG when conduced this evaluation, since it is both a GHG and Nr species?

**Response:** Thanks for your comment. N$_2$O is both a GHG and Nr species, but its environmental cost was calculated as a GHG here. This is because the cost of N$_2$O emission as Nr species is mainly to damage human health (Gu et al., 2012). But the effects of Nr losses on the direct damage costs of human health were not included in this study, which are very difficult to quantify. The environmental costs included in this study mainly refer to the global warming incurred by GHG emissions, soil acidification incurred by NH$_3$ and NO$_X$ emissions, and aquatic eutrophication caused by NH$_3$ emissions, N leaching and runoff (Xia and Yan, 2012). We have added such reasons in the methodology to make it clearer (**please see Line 208-210**).

References:

Gu, B., Ge, Y., Ren, Y., Xu, B., Luo, W., Jiang, H., Gu, B., Chang, J.: Atmospheric reactive nitrogen in China: Sources, recent trends, and damage costs, Environ. Sci. Technol., 46, 9420-9427, 2012.

Xia, Y., Yan, X.: Ecologically optimal nitrogen application rates for rice cropping in the

Taihu Lake region of China, Sustain. Sci., 7, 33-44, 2012.

5. L275-280. This discussion needs to be concise, since the effect of N fertilizer on $CH_4$ emission is beyond the focus of this study.

**Response:** Thanks for your suggestion. According to your suggestion, we have simplified the relevant discussion **(please see Line 292-294)**.

6. L289-290. The calculation of the $N_2O$ emission factor needs to be specified in the methodology.

**Response:** Thanks for your correction. According to your suggestion, we have specified the calculation of the $N_2O$ emission factor in the methodology **(please see Line 218-223)**.

7. L345. Does the straw application affect the Nr losses (e.g., $N_2O$ and $NH_3$ emission) and the subsequent calculation of Nr intensity?

**Response:** Thanks for your comment. Previous studies have proven that direct incorporation of crop straw had insignificant effects on various Nr releases (Xia et al., 2014). Because the majority of N contented in the crop straw is not easily degraded by microorganisms in a short-term period, and can be stabilized in soil in a long-term period, rather than being released as various Nr (Huang et al., 2004; Xia et al., 2014). For instance, a meta-analysis, integrating 112 scientific assessments of the crop residue incorporation on the $N_2O$ emissions, has reported that the practice exerted no statistically significant effect on the $N_2O$ releases (Shan and Yan, 2013)**.** Therefore, the effects of wheat straw incorporation on various Nr

losses were considered as negligible in this study. Moreover, previous studies have also proven that straw incorporation exerted little impacts on grain yield. For instance, a meta-analysis conducted by Singh et al. (2005) have found that incorporation of crop straw produced no significant trend in improving crop yield in rice-based cropping systems. Moreover, based on a long-term straw incorporation experiment established since 1990 in the TLR, Xia et al. (2014) have reported that long-term incorporation of wheat straw only increased the rice yield by 1%.

Therefore, in the present study, the effects of straw incorporation on NrI were considered as inappreciable. We have presented such reasons in the results and discussion part to make it clearer **(please see Line 256-263 and Line 397-406)**.

References:

Huang, Y., Zou, J., Zheng, X., Wang, Y., Xu, X.: Nitrous oxide emissions as influenced by amendment of plant residues with different C: N ratios, Soil Biol. Biochem., 36, 973-981, 2004.

Shan, J., Yan, X.Y.: Effects of crop residue returning on nitrous oxide emissions in agricultural soils, Atmos. Environ., 71, 170-175, 2013.

Singh, Y., Singh, B., Timsina, J.: Crop residue management for nutrient cycling and improving soil productivity in rice-based cropping systems in the tropics, Adv. Agron., 85, 269-407, 2005.

Xia, L., Wang, S., Yan, X.: Effects of long-term straw incorporation on the net global warming potential and the net economic benefit in a rice-wheat cropping system in China, Agric. Ecosyst. Environ., 197, 118-127, 2014.

8. L377-378. I don't think the GHGI and Nr have to have some specific relationship, although the N production and fertilization can both affect them.

**Response:** Thanks for your comment and sorry for our unclear expression. We have deleted such sentence. What we wanted to present is that extra attention should be paid to the interrelationship between the NrI and GHGI, which could provide hints for the mitigation purpose. For instance, N fertilizer production and application is an intermediate link between the NrI and GHGI (Chen et al., 2014). For the NrI, N fertilization promotes various Nr releases, exponentially or linearly (Fig.4), while N production and application made a secondary contribution to the GHGI (Table 4). Such interrelationships ought to be taken into account fully for any mitigation options pursued, in order to reduce the GHG emissions and Nr discharges from rice production simultaneously (Cui et al., 2013b; Cui et al., 2014) **(please see Line 409-416)**.

References:

Chen, X., Cui, Z., Fan, M., Vitousek, P., Zhao, M., Ma, W., Wang, Z., Zhang, W., Yan, X., Yang, J.: Producing more grain with lower environmental costs, Nature, 514, 486-489, 2014.

Cui, Z., Yue, S., Wang, G., Zhang, F., Chen, X.: In-season root-zone N management for mitigating greenhouse gas emission and reactive N losses in intensive wheat production, Environ. Sci. Technol., 47, 6015-6022, 2013b.

Cui, Z., Wang, G., Yue, S., Wu, L., Zhang, W., Zhang, F., Chen, X.: Closing the N-use efficiency gap to achieve food and environmental security, Environ. Sci. Technol., 48, 5780-5787, 2014.

9. L428. The 'ecological compensation mechanism' is a good idea to encourage famers to adopt knowledge-based agricultural managements. To make it clearer, authors need to provide more details about that rather than just giving a mention.

**Response:** Thanks for your good suggestion. According to your suggestion, we have added more details to make the 'ecological compensation mechanism' clearer **(please see Line 459-468)**.

**Reviewer 2: Some further remarks**

1. L 72, delete 'the'

**Response**: Thanks for your correction. We have revised it according to your correction (**please see Line 81**).

2. L 98-101, long sentence, needs to be split.

**Response**: Thanks for your correction. We have revised it according to your correction (**please see Line 106-109**).

3. L102, $N_2O$ should be 'nitrous oxide ($N_2O$)

**Response**: Thanks for your correction. We have revised it according to your correction (**please see Line 111**).

4. L116, delete 'an'

**Response**: Thanks for your correction. We have revised it according to your correction (**please see Line 125**).

5. L196, 'was' should be 'were'

**Response**: Thanks for your correction. We have revised it according to your correction (**please see Line 202**).

6. L230, replace 'to a reasonable rate' with 'reasonably'

**Response**: Thanks for your correction. We have revised it according to your correction (**please see Line 242**).

7. L233, delete 'without threatening food…study'

**Response**: Thanks for your correction. We have revised it according to your correction (**please see Line 245-246**).

8. L252, replace 'produced' with 'showed'

**Response**: Thanks for your correction. We have revised it according to your correction (**please see Line 265**).

9. L335, 'manufacture' should be 'production'

**Response**: Thanks for your correction. We have revised it according to your correction (**please see Line 361**).

10. L348, delete the sentence

**Response**: Thanks for your correction. We have revised it according to your correction (**please see Line 376**).

11. L427, 'has' should be 'have'

**Response**: Thanks for your correction. We have revised it according to your correction (**please see Line 460**).

12. L443, delete 'as well'

**Response**: Thanks for your correction. We have revised it according to your correction (**please see Line 479**).

13. Table 1-6, the abbreviations in the table titles should be self-explained.

**Response**: Thanks for your correction. We have revised it according to your correction (**please see the tables**).

Once again, thank you very much for your constructive comments and suggestions.

In addition, we also polished the English expressions in the whole manuscript and redrew Figure 5. All changes in the manuscript will not influence the main conclusions of the paper. And here we did not list the changes but marked in red in the attached tracked change version of manuscript. We appreciate Editor/Reviewers' warm work earnestly, and hope that the correction will meet with approval.

Yours sincerely,

XiaoyuanYan on behalf of all authors

---

## Author Comment (AC10) · 22 Jun 2016

Please see the attached file.

Please also note the supplement to this comment:
http://www.biogeosciences-discuss.net/bg-2015-620/bg-2015-620-AC10-supplement.pdf